# Change in Fatty Acid Composition in High-Temperature-Damaged Rice Grains and Its Effects on the Appearance and Physical Qualities of the Cooked Rice

**DOI:** 10.3390/foods14173097

**Published:** 2025-09-04

**Authors:** Sumiko Nakamura, Ken’ichi Ohtsubo

**Affiliations:** Faculty of Applied Life sciences, Niigata University of Pharmacy and Medical and Life Sciences, 265-1, Higashijima, Akiha-ku, Niigata City 956-8603, Japan; snaka@nupals.ac.jp

**Keywords:** eating qualities, fatty acid composition, taste degree, palmitic acid, oleic acid

## Abstract

Global warming has caused rice grains to ripen at high temperatures and become increasingly chalky, which also leads to a deterioration in the physicochemical and cooking properties of rice grains. In the present work, we first want to propose how to evaluate the palatability of rice from the high-temperature year 2022. We evaluated the qualities of 32 Japonica rice grains harvested in 2022. These showed no significant correlation with either amylose content or protein content, while the Mido score (=flavor score) showed a positive correlation with palmitic acid (r = 0.66, *p* < 0.01) and linoleic acid (r = 0.51, *p* < 0.01), in contrast to a negative correlation with oleic acid (r = −0.57, *p* < 0.01) and phosphorus content (r = −0.48, *p* < 0.01). And pasting temperatures (Pts) of polished rice flour showed significant positive correlation with the surface hardness of cooked rice grains (r = 0.53, *p* < 0.01) and significant negative correlation with their overall stickiness (r = −0.57, *p* < 0.01). In addition, Pts showed significant positive correlations with oleic acid and negative correlations with linoleic acid. Therefore, fatty acid composition could become one of the new indicators for evaluating the palatability of rice. Our second aim of this study was to determine the effects of high temperature on rice quality. It was found that oleic acid increased significantly and linoleic acid and palmitic acid decreased in 21 rice samples of the same varieties and growing regions in 2023, an abnormally hot year, compared to 2022, a normal hot year. In summary, both oleic acid content and pasting temperatures may lead to lower quality rice grains in 2023.

## 1. Introduction

Rice is one of the most important crops in the world, supplying about half the global population with calories and nutritional/bio-functional components. It can be eaten after boiling and is widely used as a material for noodles, bread, crackers, etc. The quality of rice is evaluated by sensory tests or physicochemical measurements, such as analyses of its components, pasting properties tests, and texture measurements of cooked rice [1,2].

Recently, global warming has been progressing [3], resulting in a decrease in the yield and quality of rice, which has become a severe problem all over the world [4,5]. The relationship between the night temperature and high-temperature damage [6], the mechanism of high-temperature damage [7,8], and the development of high-temperature-tolerant cultivars [9] have been studied. Juliano [10] reported that starch lipids are bound to the amylose in the endosperm starch granules as a helix between amylose and lipid hydrocarbon chains in the form of amylose–lipid complexes. Nakamura et al. reported that the fatty acid composition of Japonica rice changed when the chain length of amylopectin was altered [11]. Chalky rice grains are generated by the downregulation of starch synthase and the upregulation of starch-digesting enzymes under high temperatures during ripening [12]. Asaoka et al. [13] and Patindol and Wang [8] reported that the amylose content is lowered when rice grains ripen under high-temperature conditions, while Singh et al. [14] divided rice samples into translucent and chalky grains and compared the two groups’ proximate components, thermal properties, cooking quality, and texture when cooked. Similarly, we also reported that the surface and overall hardness of cooked rice were lower in the chalky-grain group than in the whole-grain group [15]. It remained unclear why chalky rice grains tended to retrograde rapidly, although their amylose and hardness were lower than those of the whole grains. As previously described, high-temperature damage causes low yields and quality deterioration of rice grains, which leads to worse physical properties and decreased processing suitability for rice wine [16] and rice crackers [17].

Kubo and Saio reported that the phosphorus contents in rice grains exhibited a significant positive correlation with the duration of and temperature during ripening [18]. Tabata et al. reported that a part of the phosphorus in cereal grains binds with G6 in amylopectin, while another part exists in amylose–lipid complexes [19]. Noda et al. reported that potatoes’ pasting properties become more stable by soaking them in hard water [20].

Recently, lifestyle-related diseases, such as diabetes, cardiovascular diseases, and kidney diseases, have been increasing globally [21]. As consumers are increasingly requiring their diet to contribute to maintaining and improving their health, healthy rice variants (such as brown rice, which contains a high amount of dietary fiber and vitamins; pre-germinated brown rice, which is rich in γ-aminobutyric acid (GABA); and low-protein rice, in which the protein content is reduced to less than 10%) are becoming more popular each year in Japan [22]. As the proportion of elderly people is increasing in many countries, osteoporosis due to calcium depletion has become a significant problem all over the world [23]. Although we proposed soaking and cooking chalky rice grains in hard water to improve the texture of boiled rice grains, they also contain a lot of calcium, which may be useful for preventing osteoporosis [24].

Although rice grains contain less lipids in relation to carbohydrates and proteins, lipids are important for nutrition, bio-function, and palatability [25]. Yamashita et al. reported that monoglycerides of linear saturated fatty acids bind easily with starch, but monoglycerides of unsaturated fatty acids, such as oleic acid, have more difficulty binding with starch due to the bending of molecules [26]. Morrison and Nasir reported that various rice cultivars and starches show significant positive correlations with amylose content, lipid content, pasting temperature, and pasting enthalpy [27].

Taira reported that the fatty acid compositions of Indica rice and Japonica rice are different [28]. Nga et al. also reported that Vietnamese rice’s germplasm showed variation in its fatty acid composition, and the ratio of saturated to unsaturated fatty acid in Japonica genotypes is lower than that in Indica [29]. Taira et al. compared the fatty acid compositions of rice cultivated in three cropping seasons and reported that the daily mean temperature showed a positive correlation with oleic acid and a negative correlation with linoleic acid content in japonica [30]. Gofman et al. reported that the genetic divergence affects the lipid and fatty acid composition of rice bran oil [31]. Kitta et al. measured the lipid contents and fatty acid composition of major Japanese non-glutinous rice cultivars over 4 years and proposed that the fatty acid composition was affected by the temperature during ripening [32]. Park et al. studied the effect of the milling ratio on sensory properties of cooked rice and reported that there were high negative correlations between the descriptive attributes of sweet taste, degree of agglomeration, adhesiveness, and cohesiveness of mass and the moisture, protein, and fat contents [33]. During cooking, the formation of cooked rice’s aroma is closely related to complex chemical reactions among various components, including the Maillard reaction, lipid oxidation, and thermal degradation reactions [34]. Nevertheless, there are few reports of investigations of the relationship between the fatty acid composition and high-temperature damage in rice.

It has been reported that humans can taste long-chain free fatty acids [35]. Kawai and Fushiki showed that lingual lipase decomposes triacylglycerol to glycerol and free fatty acid, which is detected as a taste substance in the olfactory system in the taste buds of the tongue [36]. Recently, lipids and their components—fatty acids—have been proven to affect the eating quality of foods as one of the taste substances, which are found along with the five gustatory substances (sweet, bitter, sour, salty, and umami) [37]. Therefore, changes in the fatty acid composition in rice grains due to high-temperature ripening may lead to a change in the taste of boiled rice grains.

The “Mido meter” was developed in 1990 by a rice milling company, Toyo Rice Co. Ltd., Japan. Shoji and Kurasawa [38] and Mizuta et al. [39] reported in academic journals in 1991 and 1996 that water is useful for evaluating rice’s palatability. From 1999 to 2001, our research project on the palatability of rice was conducted in collaboration with the Ministry of Agriculture, Forestry, and Fisheries, Japan, and we reported that the correlation of “Mido value (taste degree)” was 0.72 for various Japanese Japonica rice cultivars [40]. Following this, several breeding centers adopted the Mido meter for the selection of palatable rice cultivars [41,42], and the Mido meter is now used widely in the rice market and rice competitions [43]. Recently, the Mido meter has started to be used not only in Japan but also in China, Taiwan, and Korea [44,45].

Therefore, we used a Mido meter in addition to various physicochemical measurements to evaluate the palatability of rice.

Rivas et al. reviewed the minor compounds, extraction, advancements, and prospects of rice bran [46]. Simopoulos reported that a Western-style diet, with high amounts of ω6 unsaturated fatty acids and ω6/ω3 unsaturated fatty acids, tends to cause a range of diseases, while a diet with a high amount of ω3 unsaturated fatty acids tends to prevent various diseases [47]. Although the lipid contents in rice are not high, they do affect human health because we eat rice almost every day. Furthermore, rice bran is often used as feed [48]. Therefore, it is relevant to investigate the effect of high-temperature ripening on the fatty acid composition in rice grains.

Sato et al. [49] reported that the textural characteristic of the hardness/adhesion ratio was useful as an index for decreases in edible quality due to high temperatures during ripening, but the amylose content was not suitable. Wakamatsu et al. [50] showed that the preferred protein content of unpolished rice was estimated to be 6.0~7.0% when considering the palatability, because the proportion of white-back kernels increases when the protein content is less than 6.0%.

We investigated the relationship between the fatty acid composition and starch properties of rice grains and developed estimation formulae for oleic acid and linoleic acid contents based on the pasting property test with an RVA using 30 Japonica brown rice flours as samples [11]. In addition, we reported the estimated fatty acid composition of rice produced in 2021 using our estimation formulae [51].

Our hypotheses were that the fatty acid composition would change depending on the ripening temperature and that this composition affects the edible quality of boiled rice. In this study, we evaluated the edible qualities of 32 Japonica rice samples produced in 2022 using various physicochemical measurements, focusing especially on the relationship between Mido meter analysis and fatty acid composition. In addition, we investigated the change in fatty acid composition with the change in ripening temperature using rice samples produced in 2022 and 2023, years with extraordinarily high temperatures.

## 2. Materials and Methods

### 2.1. Materials

The unpolished rice samples harvested in Japan in 2022 (Japonica subspecies; *n* = 32) were purchased in 2023 at a local market and assessed in 2024.

The average temperature during ripening was 30.6 °C in 2023, according to the AMeDAS (Automated Meteorological Data Acquisition System, Japan), while the average temperature in 2022 was 27.5 °C [52].

The ordinary Japonica rice samples (*n* = 18) were Gohyakukawa (Fukushima prefecture), Kazesayaka (Nagano), Sasanishiki (Miyagi), Ginganoshizuku (Iwate), Hatsushimo (Aichi), Koshiibuki (Niigata), Haenuki (Yamagata), Tsugaruroman (Aomori), Aichinokaori (Aichi), Yuudai 21 (Tochigi), Akitakomachi (Ibaraki), Akitakomachi (Chiba), Akitakomachi (Akita A), Akitakom achi (Akita B), Tsuyahime (Yamagata A), Tsuyahime (Yamagata B), Tsuyahime (Shimane), and Tsuyahime (Miyagi). The high-quality premium Japonica rice samples (*n* = 10) were Koshihikari (Saga), Koshihikari (Ibaraki A), Koshihikari (Ibaraki B), Koshihikari (Shimane), Koshihikari (Niigata A), Koshihikari (Niigata B), Koshihikari (Yamagata A), Koshihikari (Yamagata B), Koshihikari (Ishikawa), and Koshihikari (Yamanashi). The low-amylose Japonica rice samples (*n* = 4) were Milky queen (Kyoto), Milky queen (Yamagata), Yumepirika (Hokkaido A), and Yumepirika (Hokkaido B). All samples were stored at 5 °C in a refrigerator.

Meanwhile, the unpolished rice samples harvested in Japan in 2023 (Japonica subspecies; *n* = 21) were purchased in 2024 at a local market and assessed in 2025. The 2022 and 2023 samples came from the same cultivars and production areas. The ordinary Japonica rice samples (*n* = 11) were Gohyakukawa (Fukushima prefecture), Kazesayaka (Nagano), Sasanishiki (Miyagi), Ginganoshizuku (Iwate), Hatsushimo (Aichi), Koshiibuki (Niigata), Haenuki (Yamagata), Tsugaruroman (Aomori), Yuudai21 (Tochigi), Tsuyahime (Yamagata A), and Tsuyahime (Shimane). The high-quality premium Japonica rice samples (*n* = 7) were Koshihikari (Saga), Koshihikari (Ibaraki A), Koshihikari (Shimane), Koshihikari (Niigata A), Koshihikari (Niigata B), Koshihikari (Yamagata A), and Koshihikari (Yamagata B). The low-amylose Japonica rice samples (*n* = 3) were Milky queen (Kyoto), Milky queen (Yamagata), and Yumepirika (Hokkaido A). All samples were stored at 5 °C in a refrigerator.

### 2.2. Measurement of the Moisture Contents of Rice Flour

The moisture contents of the polished and unpolished rice flours were measured using an oven-drying method by drying 2 g of flour samples for 1 h at 135 °C.

### 2.3. Preparation of Unpolished Rice Flour

The unpolished rice grains of the 32 rice samples from 2022 and the 21 rice samples from 2023 were pulverized to rice flour using a cyclone mill (SFC-S1; UDY, Corp., Fort Collins, CO, USA).

### 2.4. Preparation of Starch Granules

Starch granules were prepared from 7 different rice flours using the cold alkaline method [53]. Protein was removed from each sample (4 g) with 0.1% sodium hydroxide (40 mL) in a water bath with ice at 0 °C for 3 h by stirring vigorously, and the supernatant was discarded. The precipitate was washed with distilled water until it became neutral. Thereafter, the fat was removed from each precipitate with 60% ethyl alcohol (40 mL) at 0 °C for 0.5 h, and the supernatant was discarded. The defatted precipitate was then washed with acetone (40 mL) at 0 °C for 0.5 h. After discarding the supernatant, residual starch granules were dried at room temperature.

### 2.5. Iodine Absorption Spectrum

The AACs of alkali-treated rice starch were estimated using Juliano’s iodine colorimetric method [54,55]. The absorbance was measured at 620 nm (λ_max_, the peak wavelength of starch during iodine staining), which showed high correlations between the length of the glucan chain, the molecular sizes of amylose and super-long chains (SLCs) of amylopectin, and the absorbance at λ_max_ (Aλ_max_).

A degree of polymerization higher than 37% (Fb_3_) was estimated using the following Equation (1) [55]:(1)Fb_3_ (DP ≥ 37)% = 44.691 × A_λmax_ − 0.774

### 2.6. Protein Content

The nitrogen content was measured by the official AOAC 992.23. (Combustion) method for crude protein in cereal grains and oil seeds [56], using a nitrogen analyzer (Leco FP-528, LECO, St. Joseph, MI, USA). The protein content was obtained from the nitrogen content by multiplying it with the nitrogen protein conversion factor, 5.95.

### 2.7. Phosphorus Contents

The phosphorus contents of different unpolished rice samples were analyzed using the molybdenum blue method [57]. The measurement was carried out by the Japan Food Research Laboratories.

### 2.8. Measurements of Textural Properties of Boiled Rice Grains

A total of 10 g of polished rice grains was added to 16 g (1.6 times, *w*/*w*) of purified water in an aluminum cup as a control sample [58]. The physical properties of boiled rice grains were measured using the low- (25%) and high-compression (90%) methods using a Tensipresser (My Boy System, Taketomo Electric Co., Tokyo, Japan) according to the method described by Roy et al. [59]. The single-grain measurements were calculated to assess the physical properties of the cooked rice grains by measuring 20 individual grains using the following parameters: H1 for surface hardness, H2 for overall hardness, S1 for surface stickiness, S2 for overall stickiness, L3 for surface adhesion, S1/H1 (balance H1) for the stickiness-to-hardness ratio of the surface layer, S2/H2 (balance H2) for the stickiness-to-hardness ratio of the whole grain, A3/A1 (balance A1) for the adhesiveness-to-hardness ratio of the surface layer, and A6/A4 (balance A2) for the adhesiveness-to-hardness ratio of the entire grain.

### 2.9. “Mido” (=Taste Degree) of Boiled Rice

The “Mido” (=taste degree) of boiled rice grains was measured using a Mido meter (MA-90B, Toyo Rice Co., Wakayama, Japan) according to the method described by Mizuta et al. [60]. “Mido” is one of the indicators of rice palatability. Polished rice (33 g: polishing degree of 90.5%) was cooked in a water bath at 85 °C for 10 min. The cooked rice samples were kept in a vessel at room temperature for 3 min, and the reflectivity of the “water-retaining membrane” was measured by irradiating visible light through the polarizing filter. Based on the degree of transmittance and reflection of light, the “Mido” (=taste degree) was expressed using the software developed by Toyo Rice Co. Ltd., Wakayama, Japan; for example, rice No1 received 85 points, rice No2 received 68 points, and so on. Rice scoring a high “Mido” value is considered to be highly palatable [38,39,60].

### 2.10. α-Amylase Activity

The *α*-amylase activity of unpolished rice flour (*n* = 32) was determined using an enzyme assay kit (Megazyme International Ireland, Ltd., Wicklow, Ireland), with the measurement repeated three times.

### 2.11. Pasting Properties

The pasting properties of unpolished rice flours were measured using an RVA (model Super 4; Newport Scientific Pty Ltd., Warriewood, Australia). Each sample (3.5 g based on a 14% moisture content) was suspended in 25 mL of water. The measure-ment conditions were as follows: 1 min of heating at 50 °C, 4 min of heating from 50 °C to 93 °C, maintenance for 7 min at 93 °C, 4 min of cooling from 93 °C to 50 °C, and 3 min at 50 °C. The programmed heating and cooling cycle followed that of Toyoshima et al. [61].

Novel indices such as the ratio of setback to consistency (Set/Cons) (positive indi-ces of the proportion of amylopectin (DP ≥ 13)) and the ratio of maximum viscosity to final viscosity (Max/Fin) (negative indices of the proportion of amylopectin (DP ≥ 13)) were reported to be correlated very strongly with the proportion of intermediate and long chains of amylopectin: Fb1+2+3 (DP ≥ 13) [62].

### 2.12. Fatty Acid Composition

Brown rice flour (0.2 g) was added to 2 mL of hexane and mixed well, and a 2 M potassium hydroxide−methanol solution (0.2 mL) was added. After centrifugation, the concentration of the supernatant was adjusted with hexane, and fatty acid composition analyses were performed via gas–liquid chromatography (GC) (Shimadzu model GC-9A gas chromatograph with a capillary column and flame ionization detector (FID)). This method was the same as the official method (AOCS Official Method Ce2-66, 1997) [63], except for the amount of KOH-MeOH solution (0.1 mL). By using potassium hydroxide–methanol as the reaction solution, 98.6%–102% recovery was confirmed in methyl-esterification for five kinds of standard triacylglycerols: tricaprylin, trilaurin, tripalmitin, tristearin, and triolein. The reproducibility coefficients of variation (CV_R_) were sufficiently low, ranging from 0.4 to 3.2% for major fatty acids, with a peak area ratio, while those of minor fatty acids with a peak area ratio were less than 1%. The limit of detection was 0.01 g/100 g. Fatty acid composition measurements were taken for each sample by Japan Food Research Laboratories (using a gas chromatography method).

### 2.13. Statistical Analyses

We used Excel Statistics (ver. 2006; Microsoft Corp., Tokyo, Japan) for the statistical analyses of the significance of regression coefficients using Student’s *t*-test, one-way analysis of variance, and Tukey’s test. Tukey’s multiple comparison procedure was statistically analyzed using the Excel NAG Statistics add-in 2.0 (The Numerical Algorithms Group Ltd., Tokyo, Japan).

## 3. Results and Discussion

### 3.1. Iodine Absorption Spectra of 32 Japonica Rice Starch Samples from 2022

The AAC (apparent amylose content) is the most important index of the edible quality and pasting properties of rice [55]. It has been reported that the structures of rice starch are significantly influenced (up-or downregulated) by genes and the temperature during rice grain ripening [64,65]. Asaoka et al. showed that high-temperature-ripened grains contained decreased levels of amylose and long-chain amylopectin [13]. In this study, we examined whether the edible qualities of 32 Japonica rice from 2022 were influenced by high-temperature ripening.

We investigated different samples of the same cultivars—high-quality premium Japonica rice Koshihikari (*n* = 10), ordinary Japonica rice Akitakomachi (*n* = 4), Tsuyahime (*n* = 4), low-amylose Japonica Milky queen (*n* = 2), Yumepirika (*n* = 2), and other ordinary Japonica rice (*n* = 10)—according to the iodine absorption spectrum method described in our previous reports [43,44].

As shown in Table 1, the average AAC of Koshihikari (*n* = 10) was 16.5 ± 1.2%; that of Koshihikari produced in *Saga* was 15.0 ± 2.1%; in Ibaraki A, it was 14.5 ± 1.0%; that in Ibaraki B was 15.9 ± 0.0%; and in Shimane, it was 16.1 ± 0.0%, which was lower than expected. High-amylose rice starches include two types of starch functions, which are amylose and SLCs (super-long chains) of amylopectin [66,67]. It appears that the amount of super-long-chain (SLC) amylopectin was slightly increased, while some starch synthase activities were decreased by high-temperature ripening [63]. Patindol and Wang [8] and Cuili et al. [68] reported that amylose contents decrease and short chains of amylopectin increase in high-temperature-damaged rice. Our results were similar for amylose, but for amylopectin, the amount of long chains increased.

This may be due to differences between the rice cultivars and growing areas.

Similarly, the average AAC of Tsuyahime (*n* = 4) was 15.9 ± 1.3%, and that of Tsuyahime produced in Shimane was 14.2 ± 0.6%, which is very low.

While the average AAC of Akitakomachi (*n* = 4) was 15.6 ± 1.8%, that of Akitakomachi produced in Ibaraki was 14.1 ± 0.0%, and that in Chiba was 14.1 ± 0.1%, which are very low.

Similarly, the AAC of low-amylose Milky Queen (Yamagata) was 5.5 ± 1.6%, and that of ordinary Japonica rice Yuudai21 was 15.8 ± 0.4%, exhibiting a trend of lower values.

As shown in Figure 1A, the λ_max_ value tends to reflect the molecular sizes of amylose; therefore, it showed a significant positive correlation with the AAC (r = 1.00; *p* < 0.01). The Aλ_max_ value reflects not only the properties of amylose but also the amylopectin chain length [55].

The Aλ_max_ values showed significant positive correlations with Fb_3_ (DP ≥ 37) (the proportions of long chains in amylopectin) (r = 0.84; *p* < 0.01).

As shown in Figure 1B, AACs showed significant positive correlations with Fb_3_ (DP ≥ 37) (r = 0.77; *p* < 0.01). The AACs of Akitakomachi (Ibaraki) (a) and (Chiba) (b) were the same; however, the Fb_3_ (DP ≥ 37) of (Ibaraki) (a) was slightly higher than that of Akitakomachi (Chiba) (b).

As shown in Table 1 and Figure 1B, the AACs, Fb_3_ (DP ≥ 37), and Aλ_max_ values of low-amylose Japonica Milky Queen (e), (f) and Yumepirika (g), (h) were significantly different from those of the other ordinary amylose rice samples.

### 3.2. Protein and Phosphorus Contents of 32 Unpolished Rice Samples from 2022

The average temperature during ripening was 27.5 °C in 2022. Xu et al. showed that the total protein content of rice grains is increased by exposure to high temperatures during ripening [69]. Table 2 shows that the average protein content of ordinary Japonica rice was 6.1 ± 0.7%; that of Akitakomachi was 6.5 ± 0.4%; in Tsuyahimei, it was 5.9 ± 0.4%; that of Koshihikarii was 5.8 ± 0.4%; that of Milky Queen was 5.6 ± 0.6%; and in Yumepirika, it was 6.1 ± 0.4%.

Table 2 shows the phosphorus contents of the 32 unpolished Japonica rice samples from 2022. The average phosphorus content of ordinary Japonica rice was 285.0 ± 9.5 mg/100 g, with the contents varying according to the cultivar and production area. In our statistical analysis, the phosphorus contents showed significant positive correlations with long-chain fatty acids, such as arachidonic acid (r = 0.43; *p* < 0.05) and eicosenoic acid (r = 0.50; *p* < 0.01). Some phosphorus is reported to exist in phospholipids [70], and the role of phosphorus in ripening at high temperatures should be studied in the future. In a previous study, phosphorus contents exhibited a significant correlation with sunlight hours [71]; in this study, phosphorus contents exhibited a significant negative correlation with the taste degree (Mido value: r = −0.48; *p* < 0.01). As high-temperature ripening deteriorates the quality of rice, our result seems to be in accordance with a previous report by Fujisawa et al. [72].

The protein content showed a significant negative correlation with *α*-linolenic acid (r = −0.42; *p* < 0.05), and the phosphorus content showed a significant negative correlation with the taste degree (Mido value: r = −0.48; *p* < 0.01) and a positive correlation with arachidonic acid (r = 0.43; *p* < 0.05). Although we estimated that α-linolenic acid and proteins were likely to bind, other fatty acids did not show any significant correlations. We will try to clarify the reason for the aforementioned results in the future.

### 3.3. Textural Properties of Boiled Rice Grains of 32 Unpolished Japonica Rice Samples from 2022

In our previous report, the chalky grains of boiled rice showed lower hardness and a higher retrogradation degree than whole boiled grains [51].

Table 3 shows the textural properties of boiled rice grains of the 32 rice samples from 2022. As shown in Appendix A, the physical properties of the boiled rice grains were measured by the low- (25%) and high-compression (90%) methods with the Tensipresser [58]. The balance value (the ratio of stickiness to hardness) is important for evaluating the palatability of rice [58].

As shown in Table 3, there was a difference in average H1 (surface layer hardness) between Yumepirika (0.84 ± 0.04 N/cm^2^) and Milkyqueen (0.68 ± 0.02 N/cm^2^), cultivars with very low amylose contents. Other ordinary rice with slightly higher amylose contents showed lower average H1 values (0.58 ± 0.08 N/cm^2^) than those of Akitakomachi (0.76 ± 0.07 N/cm^2^), Koshihikari (0.71 ± 0.09 N/cm^2^), and Tsuyahime (0.82 ± 0.04 N/cm^2^), which also have low amylose contents. The characteristics of the boiled rice grains may have been affected by the change in the fine structure of amylopectin due to high-temperature ripening. Moreover, H2 (overall hardness) showed a similar tendency to H1.

The average S1 (surface layer stickiness) value of Yumepirika and Tsuyahime was 0.19 ± 0.02 N/cm^2^; that of Koshihikari and Milky Queen was 0.15 ± 0.02 N/cm^2^; in Akitakomachi, it was 0.14 ± 0.04 N/cm^2^; and that of the other ordinary rice varieties was 0.11 ± 0.03 N/cm^2^. The average S2 (overall stickiness) showed a similar tendency to S1 (surface layer stickiness).

As a result, the H1 of boiled rice grains showed significant positive correlations with H2 (r = 0.74; *p* < 0.01), S1 (r = 0.82; *p* < 0.01), and S2 (r = 0.71; *p* < 0.01). Moreover, the H2 of boiled rice grains showed significant positive correlations with S1 (r = 0.55; *p* < 0.01) and S2 (r = 0.65; *p* < 0.01). The chalky grains of boiled rice seemed to exhibit lower hardness and stickiness levels than the whole grains. The texture of boiled rice changes depending on a number of factors, such as amylose content and amylopectin microstructures. As the amylopectins of rice grains ripen under high temperatures, they have more long-chain glucans, making the boiled rice grains harder and less sticky [73]. Furthermore, chalky grains tend to retrograde more rapidly than whole grains [51].

As shown in Figure 2, the balance A2 values (A6/A4) (overall balance degree) of the low-amylose varieties Milky Queen (0.17 ± 0.01) and Yumepirika (0.16 ± 0.01) were higher than those of Tsuyahime (0.14 ±0.00), Koshihikari (0.12 ± 0.01), Akitakomachi (0.12 ± 0.02), and the other ordinary rice samples (0.12 ± 0.03). The balance A2 value (=A6/A4) of boiled rice grains showed a significant negative correlation with Fb3 (DP ≥ 37) (r = −0.62; *p* < 0.01).

As shown in Figure 3, the formula presented below was obtained using the 32 rice samples from 2022 and used to estimate balance A2 (A6/A4) (overall balance degree), which was based on Fb_3_ (DP ≥ 37), linoleic acid (18:2*n*−6), and α-amylase activity. The equation yielded a multiple regression coefficient of 0.64**.(2)A2 (A6/A4) (overall balance degree) = −0.016 × Fb_3_ (DP ≥ 37) −0.001 × linoleic acid −0.110 × α-amylase activity

It appears that the long chains of amylopectins and α-amylase activities were increased by high-temperature ripening, although the contents of palmitic acid, linoleic acid, and α-linolenic acid decreased, which caused a harder and less sticky texture in the boiled rice grains.

The H2 (overall hardness) of the boiled rice grains showed a significant positive correlation with palmitoleic acid (16:1) (r = 0.45; *p* < 0.01), and S2 showed a significant positive correlation with palmitoleic acid (16:1) (r = 0.44; *p* < 0.05). Moreover, the L3 (surface adhesion) of the boiled rice grains showed a significant positive correlation with α-linolenic acid (r = 0.49; *p* < 0.01). As a result, it appears that fatty acid is an indicator of the taste degree of high-temperature-damaged rice.

### 3.4. Taste Degree of Boiled Rice of 32 Polished Japonica Rice Samples from 2022

As shown in Appendix A, the “Mido” (=taste degree) of the boiled rice grains was measured using a Mido meter according to the method described by Mizuta et al. [39]. “Mido” is an indicator of rice palatability. Based on the degree of transmittance and reflection of light, “Mido” (=taste degree) was expressed using the software developed by Toyo Rice Co., Ltd., Wakayama, Japan; for example, rice No1 received 85 points, rice No2 received 68 points, and so on. Rice scoring a high “Mido” value seems to be highly palatable [40,41,42].

Shoji and Kurasawa [38] used 25 rice samples produced in Fukushima prefecture and reported that the results of their sensory test showed high correlation (r = 0.996) with Mido values (=taste degree). Mizuta et al. compared the results of a sensory test and Mido meter measurements using rice samples cultivated in Miyagi prefecture for 4 years and reported that they showed a high positive correlation (r = 0.692) [39]. Sato et al. examined the possibility of using a Mido meter and RVA for the efficient selection of highly palatable lines in rice breeding and concluded that the two methods were useful [49]. Itayagoshi et al. used 21 rice samples produced in 2017 as samples, carried out a sensory test and physicochemical measurements, and reported that the Mido value showed a significant positive correlation (r = 0.611) with the results of the sensory test [73].

As shown in Figure 4, the average Mido value of Tsuyahime was 78.7 ± 7.0; that of low-amylose rice was 74.3 ± 5.6; in other ordinary rice it was 72.4 ± 3.1; that of Koshihikari was 70.6 ± 7.2; and that of Akitakomachi was 66.9 ± 7.0. The Mido values of rice samples grown in Yamagata and Niigata seemed to be slightly higher.

### 3.5. α-Amylase Activities of 32 Polished Japonica Rice Samples from 2022

The α-amylase activities of the chalky rice grains showed higher values than those of whole rice grains, while the values of protease and xylanase activities showed similar tendencies [12,74,75].

Appendix A shows the α-amylase activities of the 32 unpolished rice samples from 2022. As shown in Appendix A, the average α-amylase activity of Yumepirika was 0.121 ± 0.014 CU/g; that of Tsuyahime was 0.090 ± 0.026 CU/g; in Akitakomachi, it was 0.114 ± 0.010 CU/g; that of Koshihikari was 0.079 ± 0.016 CU/g; that of Milkyqueen was 0.099 ± 0.006 CU/g; and for the other ordinary rice, it was 0.079 ± 0.021 CU/g. It seemed that rice samples grown in Ibaraki, Akita, and Hokkaido were more severely damaged by high-temperature ripening.

### 3.6. Pasting Properties of 32 Unpolished Rice Samples from 2022 and 21 Rice Samples from 2023

Global warming and other environmental trends are impacting rice production. Rice producers all over the world need to understand the implications of environmental changes. The chalky rice grains showed lower pasting viscosity values, such as Max. vis and Final. vis., than the whole grains; in contrast, they showed higher *α*-amylase activities than the whole grains [71]. Pasting is one of the most important rheological indicators of the cooking quality or processing suitability of rice starch [76].

The lipid contents and fatty acid composition of rice were affected by the ripening temperature [28,30]. We developed novel estimation formulae for oleic acid and linoleic acid contents based on pasting properties [11].

In our previous paper, the pasting temperature of high-temperature-ripened rice was tested using an RVA (rapid visco analyzer) and showed a positive correlation with retrogradation of the degree of hardness, both H1 (RD) (retrogradation of the degree of hardness of the surface layer) and H2 (RD) (retrogradation of the degree of hardness of the entire grain) after cooking, because their starches included super-long-chain (SLC) amylopectin [77,78]. We found that chalky rice grains are characterized by high α-amylase activities, high protease activity, and high xylanase activity; therefore, the pasting properties of the tested samples showed lower values, such as Max.vis (maximum viscosity) and Final. vis (final viscosity). In contrast, the Pt values (pasting temperature) of these samples were slightly higher than those of whole grains [61]. Itayagoshi et al. [73] reported that when measuring with an RVA, the pasting temperatures showed a significant negative correlation with the overall evaluation in sensory tests.

Table 4 shows the pasting properties of the 32 unpolished Japonica rice from 2022.

Figure 5 shows that the Pt values of Akitakomachi (68.4 ± 2.6) °C, Koshihikari (68.3 ± 1.4) °C, and Tsuyahime (68.2 ± 2.1) °C were higher than those of low-amylose rice (66.5 ± 1.2) °C and another ordinary Japonica rice sample (60.7 ± 2.4) °C. It seemed that the Akitakomachi, Koshihikari, and Tsuyahime varieties were damaged more severely by high temperature, and those produced in Ibaraki, Chiba, and Shimane prefectures showed particularly high Pt values.

Zhang et al. [64] reported that the chain length distribution of amylopectin determined the gelatinization temperature of starch, as well as its enthalpy changes and pasting properties. In our previous paper, cultivars with a high proportion of long-chain amylopectin showed a high Pt and higher contents of USF (unsaturated fatty acid, eicosenoic acid) and SF (arachidic acid, behenic acid, lignoceric acid), while samples with a low proportion of long-chain amylopectin showed low Pt and Cons (consistency) values and high linoleic acid and linolenic acid contents [11].

As shown in Table 5, Pt showed significant positive correlations with oleic acid (r = 0.46; *p* < 0.01) and arachidic acid (r = 0.57; *p* < 0.01) and significant negative correlations with palmitic acid (r = −0.46; *p* < 0.01) and linoleic acid (r = −0.56; *p* < 0.01). It appears that the long amylopectin chain, which binds with fatty acids such as oleic acid and arachidic acid, is increased during high-temperature ripening. More oleic acid may bind with long amylopectin chains, which may lead to an increased pasting temperature and a lower taste degree. This mechanism may be related to the tendency of Pt contents to rise with increases in long chains in high-Pt rice cultivars, such as indica high-amylose rice [29].

Table 6 shows the pasting properties of 21 unpolished rice samples from 2023, of which the cultivars and production areas were the same as those produced in 2022. The average temperature during ripening was 30.6 °C in 2023, while it was 27.5 °C in 2022.

Figure 6 shows that the average Pt of the 21 unpolished rice from 2023 was 69.7 ± 1.3 °C, which was significantly higher than that of the rice from 2022 (64.8 ± 4.3) °C.

Figure 7A reveals that the Pt of the 21 unpolished rice samples from 2022 showed a significant positive correlation with oleic acid (r = 0.35; *p* < 0.01). In contrast, Figure 7B shows that rice harvested in 2023 showed a lower (r = 0.49) correlation. The Pt, amount of long chains in amylopectin, and oleic acid content all seemed to be increased by high-temperature ripening.

### 3.7. Fatty Acid Compositions of 32 Unpolished Rice Samples from 2022 and 21 Unpolished Rice Samples from 2023

Lipids are less abundant in rice than carbohydrates and proteins; however, their functional properties are important [79]. Park et al. studied the effect of the milling ratio on sensory properties of cooked rice and demonstrated a correlation between descriptive attributes relating to sweet taste and fat contents [33]. Cameron and Wang reported that the crude lipid content was negatively correlated with the hardness of cooked rice and positively correlated with their stickiness [79]. Lipids in the rice endosperm are always located in a compound with starch granules and interfere with amylose molecules and linear regions of amylopectin to form inclusion complexes [80]. Yoon et al. reported that oleic acid concentrations were higher in both the highly palatable Koshihikari and less palatable Singeumo than in Dongjin, whereas Dongjin showed high levels of palmitic and linoleic acids [81]. Guo et al. proposed a positive association between the fat content and edible quality of rice using 94 homozygous recombinant inbred lines (RILs) [82]. Zhang et al. reported that natural variation in the fatty acid desaturase gene affects the linolenic acid content and starch pasting viscosity in rice grains [83].

PUFAs (polyunsaturated fatty acids) such as linoleic acid (18: 2*n*−6), *α*-linoleic acid (18: 3*n*−3), arachidonic acid, EPA (eicosapentaenoic acid), and DHA (docosahexaenoic acid) are essential fatty acids, and these PUFAs cannot be synthesized negatively. Nevertheless, arachidonic acid can be synthesized from linoleic acid and is one of the major fatty acids in the brain, along with DHA [84,85]. Arachidonic acid is contained in breast milk and is important for infant nutrition. Dietary guidelines always center around compound starch granules, and a deficiency of or interference with essential fatty acids may play an important role in the development of coronary heart disease, particularly during the early growth period [86]. Fatty acids with high n6/n3 ratios have been reported to increase the risk of cardiovascular disease, cancer, and inflammatory, autoimmune, and other chronic diseases, while those with low n6/n3 ratios tend to exert suppressive effects [87,88]. Ibrahim et al. reported that changing the dietary n6/n3 ratio using different oil sources affected the performance, behavior, cytokines, mRNA expression, and meat fatty acid profile of broiler chickens [83].

The lipid contents and fatty acid composition of rice were affected by the ripening temperature [28,30].

The relation between the palatability of rice and its mido (taste) value (taste degree) was examined to efficiently select highly palatable lines in rice breeding. This would make the Mido value useful for selecting highly palatable rice lines [38,39,40,41,42,43].

Table 7 shows the fatty acid compositions of the 32 unpolished Japonica rice samples from 2022. As shown in Table 7, the average palmitic acid content of Yumepirika was 23.9 ± 0.1%; that of Koshihikari was 22.4 ± 0.8%; in Tsuyahime, it was 22.3 ± 0.8%; that of other ordinary rice was 22.5 ± 0.6%; in Milkyqueen, it was 21.8 ± 1.6%; and that of Akitakomachi was 21.0 ± 1.6%. The average contents of linoleic acid and *α*-linolenic acid showed similar tendencies. In contrast, the average content of oleic acid in Akitakomachi was 40.5 ± 2.9%; that of Tsuyahime was 37.4 ± 1.3%; that of Koshihikari was 36.8 ± 1.3%; in Milky queen, it was 36.8 ± 1.6%; in other ordinary rice, it was 36.6 ± 1.3%; and that of Yumepirika was 34.4 ± 0.4%. The Akitakomachi samples showed low ratios of palmitic acid, linoleic acid, and *α*-linolenic acid and a high ratio of oleic acid. It appears that Akitakomachi samples were affected by high-temperature ripening.

Figure 8 shows that the taste degree of the 32 unpolished Japonica rice samples from 2022 showed a positive correlation with Palmitic acid (r = 0.66; *p* < 0.01) and a negative correlation with Oleic acid (r = −0.57; *p* < 0.01). It seemed that the oleic acid content was increased by high-temperature ripening, while the taste degree was decreased.

We carried out principal component analysis using all the physicochemical measurements as variables (Figure 9). The first principal component (contribution rate: 77.5%) consisted of λ_max_/A_max_, balance A2 (A6/A4), etc. (positive), and Fb_3_ (DP ≥ 37), Aλ_max_, etc. (negative), which mainly related to the amylose and amylopectin contents of starch components. The second principal component (contribution rate: 14.0%) consisted of BD, Max. vis, myristic acid, etc. (positive), and SB (setback), palmitic acid, α-amylase activities, etc. (negative), which mainly related to the pasting property, fatty acid composition, and amylolytic enzymes. As a result, the first principal component was mainly affected by the starch microstructure, and the second principal component was mainly affected by the binding of fatty acids with starch and hydrolytic enzyme activities. The second principal component, which includes fatty acids and amylolytic enzymes, was found to make the main contribution to the overall edible quality of boiled rice through the starch microstructure. It also seemed that Tsuyahime and Koshihikari did not fluctuate significantly, while in contrast, Akitakomachi, ordinary Japonica rice, and low-amylose rice varied noticeably.

Table 8 shows a comparison of the fatty acid compositions of 21 unpolished rice samples, harvested in 2022 and 2023. The ripening temperature in 2023 was 3 °C higher than in 2022. As a result, Oleic acid (38.0 ± 1.5)% was significantly higher in 2023 than in 2022 (36.5 ± 1.4)%, at the level of 1%. In contrast, Linoleic acid (33.4 ± 0.9)% and *α*-linoleic acid (1.2 ± 0.1)% were significantly lower in 2023 than in 2022 (34.8 ± 0.8% and 1.3 ± 0.1%), respectively, at the level of 1%, while Palmitic acid showed a similar tendency in 2023 (21.9 ± 0.8)% to that in 2022 (22.5 ± 0.9)% at the level of 5%. Although we reported the fatty acid composition of rice produced in 2016, a year with ordinary temperatures [11], Table 8 shows the difference in fatty acid compositions of rice produced under high temperatures (2022) and extremely high temperatures (2023). We carried out cluster analysis of the 32 samples from 2023 using Pt, taste degree, oleic acid, linoleic acid, palmitic acid, and Fb_3_ (37 ≥ DP) as variables. The results are shown in Appendix A.

The bio-functionality and physicochemical quality of rice grains are very important. An excessive amount of n6 PUFAs and a very high n6/n3 ratio promote the pathogenesis of several diseases [47]. Antunes et al. compared the ratio of myristic acid to DHA and the ratio of n6 to n3 as biomarkers for nonalcoholic fatty liver disease (NAFLD) [87].

Figure 10 shows the ratios of n6 (omega-6) fatty acid to n3 (omega-3) for 32 rice cultivars produced in 2022. The average ratios (n6/n3) of Akitakomachi samples were higher than those of the other rice samples. It appears that Akitakomachi was damaged severely by high-temperature ripening. The (n6/n3) ratios in 2023 (28.9 ± 2.8) were higher than in 2022 (27.7 ± 2.5), which indicates that the ripening temperature affected these ratios. This may not be a favorable change in terms of bio-functionality, not only for humans [85,87] but also for animals when rice bran is used as a material for cattle feed [88].

## 4. Conclusions

In the present study, we investigated the effects of ripening at high temperatures on changes in the fatty acid content of rice grains. We also evaluated the eating quality of rice grains using various physicochemical measurements, focusing on the relationship between the flavor grade of boiled rice grains and fatty acid content. We evaluated 32 japonica rice grains harvested in 2022.

Their eating qualities (taste degrees) showed significant positive correlations with palmitic acid (r = 0.66, *p* < 0.01) and linoleic acid (r = 0.51, *p* < 0.01), in contrast, a significant negative correlation with oleic acid (r = −0.57, *p* < 0.01) and phosphorus content (r = −0.48, *p* < 0.01). Although it was reported that the texture of boiled rice grains was affected by the contents of amylose and protein, we found that the texture of high-temperature ripened rice grains is mainly influenced by starch microstructure, fatty acid composition, and amylase activity.

From our results, Koshihikari and Tsuyahime are promising high-temperature-tolerant rice varieties. It should be kept in mind that these results only apply to our rice sample, 32 Japonica rice varieties produced in Japan. It would be necessary to use a wider range of rice samples, including japonica rice from other countries or indica rice samples, if we want to further elaborate on our conclusions.

The comparison between 2022 and 2023 using 21 rice grains of the same rice varieties and the same growing areas showed that the oleic acid content was higher in the 2023 rice grains, and pasting temperatures were higher than in the 2022 rice grains, which would have resulted in a lower eating quality of the 2023 rice grains. Although our results were not sensory evaluated, this result would be useful for breeding high-temperature-tolerant, tasty, and healthy rice varieties.

## Figures and Tables

**Figure 1 foods-14-03097-f001:**
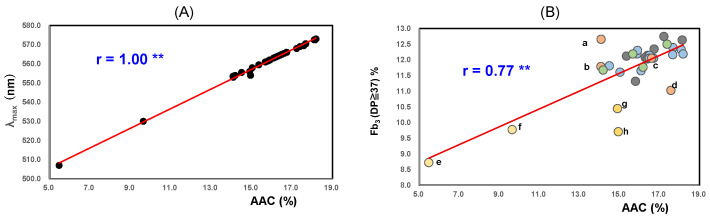
Correlation between AAC and λ_max_ (**A**), Fb_3_ (DP ≥ 37%) (**B**) of 32 Japonica rice starches from 2022. ** Correlation is significant at 1% by the method of Tukey’s multiple comparison. a; Akitakomachi (Ibaraki); b; Akitakomachi (Chiba); c; Akitakomachi (Akita) A; d; Akitakomachi (Akita) B; e; Milkyqueen (Yamagata); f; Milkyqueen (Kyoto); g; Yumepirika (Hokkaido) B; h; Yumepirik (Hokkaido) A. (**A**), Relationship between AAC and λ_max_ in high-temperature-ripened rice from 2022. (**B**), Relationship between AAC and Fb_3_ (DP ≥ 37%) in high-temperature-ripened rice from 2022.

**Figure 2 foods-14-03097-f002:**
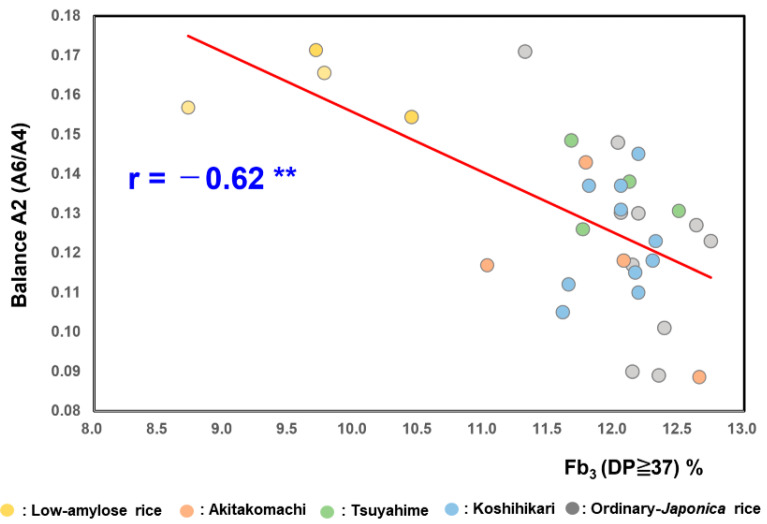
Correlation between balance A2 (A6/A4) of boiled rice and Fb3 (DP ≥ 37) of starches in 32 rice samples from 2022. ** Correlation is significant at 1% by the method of Tukey’s multiple comparison.

**Figure 3 foods-14-03097-f003:**
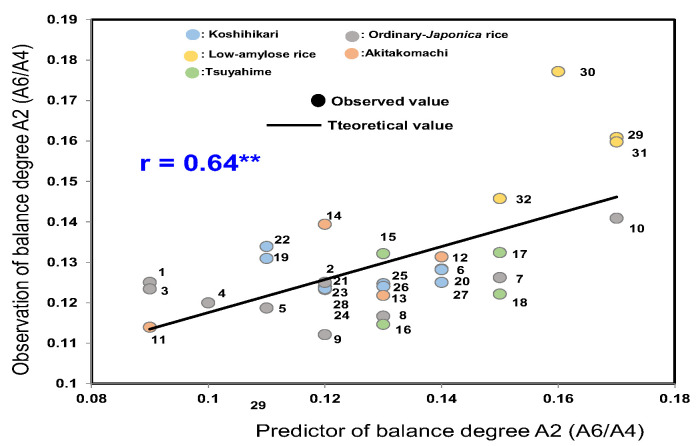
Formulae for estimating the balance degree A2 (A6/A4) based on Fb_3_ (DP ≥ 37), linoleic acid (18: 2*n*−6), and α-amylase activity. 1, Gohyakukawa; 2, Kazesayaka; 3, Sasanishiki; 4, Ginganosizuku; 5, Hatsushimo; 6, Koshiibuki; 7, Haenuki; 8, Tsugaruroman; 9, Aichinokaori; 10, Yuudai21; 11, Akitakomachi (Ibaraki); 12, Akitakomachi (Chiba); 13, Akitakomachi (Akita A); 14, Akitakomachi (Akita B); 15, Tsuyahime (Yamagata A); 16, Tsuyahime (Yamagata B); 17, Tsuyahime (Shimane); 18, Tsuyahime (Miyagi); 19, Koshihikari (Saga); 20, Koshihikari (Ibaraki A); 21, Koshihikari (Ibaraki B); 22, Koshihikari (Shimane); 23, Koshihikari (Niigata A); 24, Koshihikari (Niigata B); 25, Koshihikari (Yamagata A); 26, Koshihikari (Yamagata B); 27, Koshihikari (Ishikawa); 28, Koshihikari (Yamanashi); 29, Milkyqueen (Kyoto); 30, Milkyqueen (Yamagata); 31, Yumepirika (Hokkaido A); 32, Yumepirika (Hokkaido B).

**Figure 4 foods-14-03097-f004:**
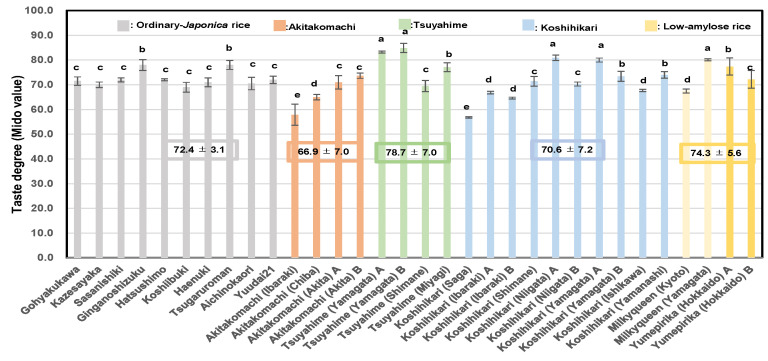
Analyses of taste degree (Mido) of 32 rice samples from 2022 using a Mido meter. Different letters (a, b, c, d, e) indicate significant difference.

**Figure 5 foods-14-03097-f005:**
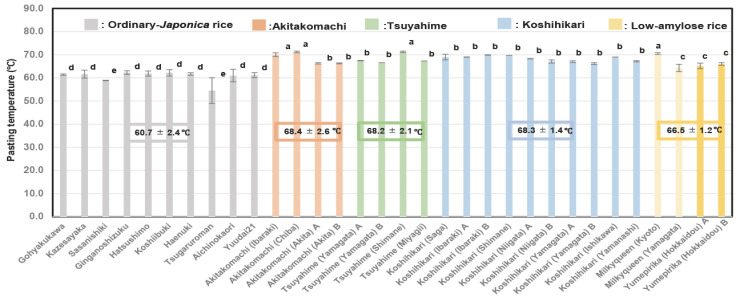
Pasting temperatures of different unpolished rice varieties based on 32 rice samples from 2022. Different letters (a, b, c, d, e) indicate significant differences. Correlation is significant at 1% based on Tukey’s multiple comparisons method.

**Figure 6 foods-14-03097-f006:**
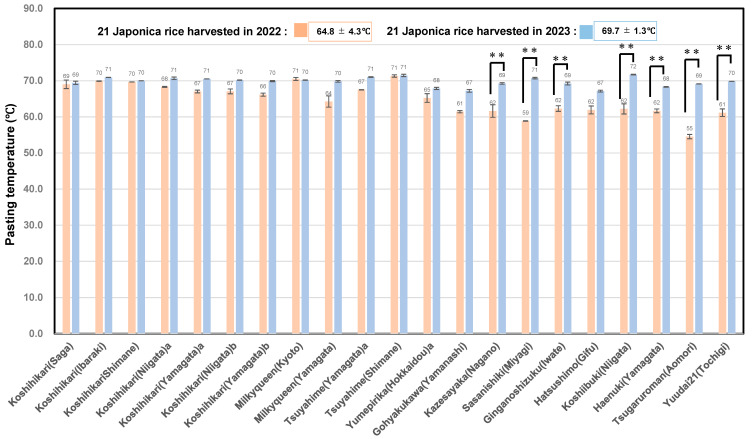
Pasting temperatures of 21 rice samples harvested in 2022 and 2023. ** Correlation is significant at 1% based on Tukey’s multiple comparison method.

**Figure 7 foods-14-03097-f007:**
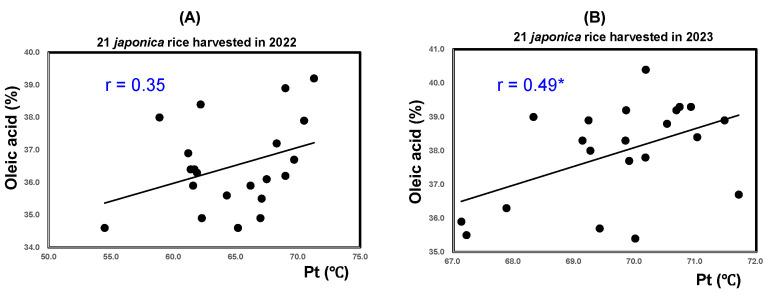
Correlation between results of analyses of pasting temperature (Pt) and oleic acid of 21 varieties of unpolished rice harvested in 2022 and 2023. * Correlation is significant at 5% based on Tukey’s multiple comparison method.

**Figure 8 foods-14-03097-f008:**
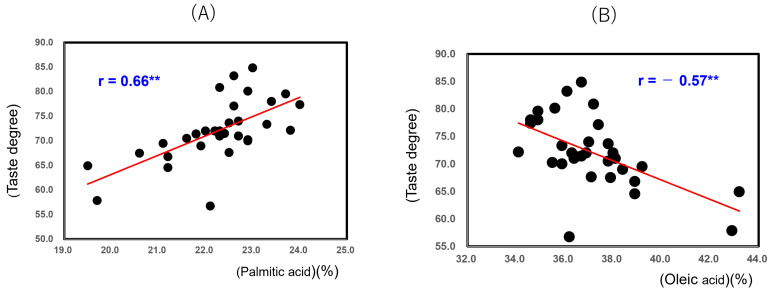
Correlation between taste degree (Mido values) and palmitic acid (**A**), and oleic acid (**B**) of 32 unpolished Japonica rice samples from 2022. ** Correlation is significant at 1% by the method of Tukey’s multiple comparison.

**Figure 9 foods-14-03097-f009:**
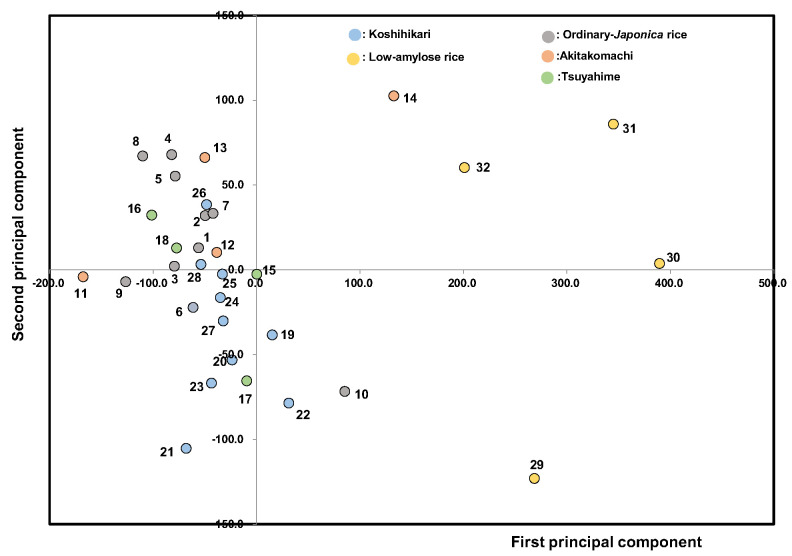
Principal component analysis of 32 unpolished Japonica rice samples from 2022. 1, Gohyakukawa; 2, Kazesayaka; 3, Sasanishiki; 4, Ginganosizuku; 5, Hatsushimo; 6, Koshiibuki; 7, Haenuki; 8, Tsugaruroman; 9, Aichinokaori; 10, Yuudai21; 11, Akitakomachi (Ibaraki); 12, Akitakomachi (Chiba); 13, Akitakomachi (Akita A); 14, Akitakomachi (Akita B); 15, Tsuyahime (Yamagata A); 16, Tsuyahime (Yamagata B); 17, Tsuyahime (Shimane); 18, Tsuyahime (Miyagi); 19, Koshihikari (Saga); 20, Koshihikari (Ibaraki A); 21, Koshihikari (Ibaraki B); 22, Koshihikari (Shimane); 23, Koshihikari (Niigata A); 24, Koshihikari (Niigata B); 25, Koshihikari (Yamagata A); 26, Koshihikari (Yamagata B); 27, Koshihikari (Ishikawa); 28, Koshihikari (Yamanashi); 29, Milky queen (Kyoto); 30, Milky queen (Yamagata); 31, Yumepirika (Hokkaido A); 32, Yumepirika (Hokkaido B).

**Figure 10 foods-14-03097-f010:**
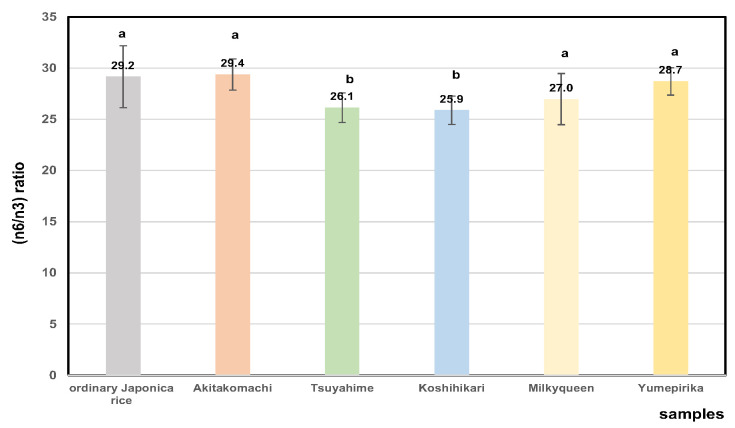
Comparison of n6/n3 (omega-6/omega-3) ratios of fatty acid of Japonica 32 unpolished rice in 2022. Different letters (a, b) indicate significant differences.

**Table 1 foods-14-03097-t001:** Analyses of iodine absorption curves of 32 Japonica rice starch samples from 2022.

	AAC	λ_max_	A_λmax_	Fb_3_(37 > DP)	λ_max_/A_λmax_
	(%)	(nm)		(%)	
**Gohyakukawa**	**16.4 ± 0.7 b**	**564.5 ± 3.5 b**	**0.289 ± 0.003 a**	**12.1 ± 0.1 b**	**1953.3 ± 6.9 c**
**Kazesayaka**	**16.3 ± 0.6 b**	**564.0 ± 2.8 b**	**0.288 ± 0.002 a**	**12.1 ± 0.1 b**	**1961.8 ± 4.6 c**
**Sasanishiki**	**16.7 ± 0.6 b**	**566.0 ± 2.8 b**	**0.294 ± 0.002 a**	**12.3 ± 0.1 b**	**1928.5 ± 4.3 d**
**Ginganoshizuku**	**17.7 ± 0.2 a**	**570.5 ± 0.7 a**	**0.295 ± 0.001 a**	**12.4 ± 0.0 b**	**1937.2 ± 2.3 d**
**Hatsushimo**	**15.9 ± 0.3 b**	**562.0 ± 1.4 b**	**0.220 ± 0.000 d**	**12.2 ± 0.0 b**	**1937.9 ± 4.9 d**
**Koshiibuki**	**15.4 ± 0.2 c**	**559.5 ± 0.7 c**	**0.289 ± 0.002 a**	**12.1 ± 0.1 b**	**1939.4 ± 11.8 d**
**Haenuki**	**16.4 ± 0.4 b**	**564.5 ± 2.1 b**	**0.287 ± 0.002 a**	**12.0 ± 0.1 b**	**1970.4 ± 7.2 c**
**Tsugaruroman**	**18.1 ± 0.4 a**	**572.5 ± 2.1 a**	**0.300 ± 0.001 a**	**12.6 ± 0.1 a**	**1908.4 ± 16.1 e**
**Aichinokaori**	**17.2 ± 0.6 b**	**568.0 ± 2.8 b**	**0.303 ± 0.001 a**	**12.7 ± 0.0 a**	**1877.7 ± 13.7 f**
**Yuudai21**	**15.8 ± 0.4 b**	**561.5 ± 2.1 b**	**0.271 ± 0.001 b**	**11.3 ± 0.0 b**	**2075.8 ± 2.4 b**
**Akitakomachi (Ibaraki)**	**14.1 ± 0.0 c**	**553.0 ± 0.0 c**	**0.301 ± 0.002 a**	**12.7 ± 0.1 a**	**1840.3 ± 13.0 f**
**Akitakomachi (Chiba)**	**14.1 ± 0.1 c**	**553.5 ± 0.7 c**	**0.281 ± 0.001 b**	**11.8 ± 0.1 b**	**1969.8 ± 12.4 c**
**Akitakomachi (Akita) A**	**16.6 ± 0.1 b**	**565.5 ± 0.7 b**	**0.287 ± 0.004 a**	**12.1 ± 0.2 a**	**1970.6 ± 31.6 c**
**Akitakomachi (Akita) B**	**17.6 ± 0.2 a**	**569.5 ± 0.7 a**	**0.264 ± 0.000 c**	**11.0 ± 0.0 b**	**2157.2 ± 2.7 b**
**Tsuyahime (Yamagata) A**	**16.2 ± 0.2 b**	**563.5 ± 0.7 b**	**0.281 ± 0.001 b**	**11.8 ± 0.0 b**	**2008.9 ± 7.6 b**
**Tsuyahime (Yamagata) B**	**17.4 ± 0.3 b**	**569.0 ± 1.4 a**	**0.297 ± 0.001 a**	**12.5 ± 0.1 a**	**1915.9 ± 13.9 e**
**Tsuyahime (Shimane)**	**14.2 ± 0.6 c**	**554.0 ± 2.8 c**	**0.279 ± 0.002 b**	**11.7 ± 0.1 b**	**1989.3 ± 25.3 b**
**Tsuyahime (Miyagii)**	**15.7 ± 0.3 b**	**561.0 ± 1.4 b**	**0.290 ± 0.000 a**	**12.2 ± 0.0 a**	**1934.5 ± 4.9 d**
**Koshihikari (Saga)**	**15.0 ± 2.1 c**	**558.0 ± 9.9 c**	**0.277 ± 0.001 b**	**11.6 ± 0.1 b**	**2014.4 ± 25.5 b**
**Koshihikari (Ibaraki) A**	**14.5 ± 1.0 c**	**555.5 ± 4.9 c**	**0.282 ± 0.002 b**	**11.8 ± 0.1 b**	**1973.5 ± 32.5 c**
**Koshihikari (Ibaraki) B**	**15.9 ± 0.0 b**	**562.0 ± 0.0 b**	**0.293 ± 0.008 a**	**12.3 ± 0.3 b**	**1922.0 ± 51.1 d**
**Koshihikari (Shimane)**	**16.1 ± 0.0 b**	**563.0 ± 0.0 b**	**0.278 ± 0.000 b**	**11.7 ± 0.0 b**	**2025.2 ± 0.0 b**
**Koshihikari (Niigata) A**	**18.1 ± 0.2 a**	**572.5 ± 0.7 a**	**0.293 ± 0.001 a**	**12.3 ± 0.1 b**	**1953.9 ± 7.0 c**
**Koshihikari (Niigata) B**	**17.7 ± 1.3 a**	**570.5 ± 6.4 a**	**0.290 ± 0.004 a**	**12.2 ± 0.2 b**	**1970.9 ± 46.1 c**
**Koshihikari (Yamagata) A**	**18.2 ± 0.3 a**	**573.0 ± 1.4 a**	**0.290 ± 0.003 a**	**12.2 ± 0.1 b**	**1975.9 ± 14.4 c**
**Koshihikari (Yamagata) B**	**16.5 ± 0.3 b**	**565.0 ± 1.4 b**	**0.287 ± 0.000 a**	**12.1 ± 0.0 b**	**1968.6 ± 4.9 c**
**Koshihikari (Ishikawa)**	**16.7 ± 0.6 b**	**566.0 ± 2.8 b**	**0.287 ± 0.001 a**	**12.1 ± 0.1 b**	**1972.1 ± 0.1 c**
**Koshihikari (Yamanashi)**	**16.5 ± 1.5 b**	**565.0 ± 7.1 b**	**0.289 ± 0.003 a**	**12.1 ± 0.1 b**	**1955.2 ± 43.6 c**
**Milkyqueen (Kyoto)**	**9.6 ± 0.6 d**	**530.0 ± 2.8 d**	**0.236 ± 0.001 d**	**9.8 ± 0.1 c**	**2245.8 ± 1.5 a**
**Milkyqueen (Yamagata)**	**5.5 ± 1.6 e**	**507.0 ± 7.1 e**	**0.213 ± 0.001 d**	**8.7 ± 0.0 d**	**2386.0 ± 41.2 a**
**Yumepirika (Hokkaidou) A**	**14.9 ± 0.5 c**	**554.0 ± 2.8 c**	**0.235 ± 0.002 d**	**9.7 ± 0.1 c**	**2362.5 ± 9.3 a**
**Yumepirika (Hokkaidou) B**	**14.9 ± 0.3 c**	**556.0 ± 1.4 c**	**0.251 ± 0.001 c**	**10.4 ± 0.1 c**	**2215.2 ± 18.1 a**

Different letters (a, b, c, d, e, f) indicate significant difference.

**Table 2 foods-14-03097-t002:** Analyses of protein and phosphorus contents of the 32 unpolished rice samples from 2022.

	Protein	Phosphorus
		Contents
	(%)	(mg/100 g)
**Gohyakukawa**	**5.8 ± 0.0 c**	**290.0 ± 0.5 c**
**Kazesayaka**	**5.7 ± 0.0 c**	**286.0 ± 0.6 d**
**Sasanishiki**	**4.6 ± 0.0 d**	**280.0 ± 0.5 d**
**Ginganoshizuku**	**6.3 ± 0.0 b**	**283.0 ± 1.1 d**
**Hatsushimo**	**6.6 ± 0.1 b**	**270.0 ± 0.8 e**
**Koshiibuki**	**5.9 ± 0.0 c**	**300.0 ± 0.6 c**
**Haenuki**	**6.1 ± 0.0 c**	**278.0 ± 0.5 d**
**Tsugaruroman**	**6.4 ± 0.0 b**	**284.0 ± 0.8 d**
**Aichinokaori**	**7.3 ± 0.1 a**	**300.0 ± 1.1 c**
**Yuudai21**	**6.1 ± 0.0 c**	**279.0 ± 0.6 d**
**Akitakomachi (Ibaraki)**	**7.1 ± 0.0 a**	**306.0 ± 1.2 c**
**Akitakomachi (Chiba)**	**6.4 ± 0.0 b**	**330.0 ± 2.0 b**
**Akitakomachi (Akita) A**	**6.2 ± 0.0 b**	**276.0 ± 1.1 d**
**Akitakomachi (Akita) B**	**6.4 ± 0.0 b**	**283.0 ± 0.8 d**
**Tsuyahime (Yamagata) A**	**6.1 ± 0.1 c**	**275.0 ± 1.2 d**
**Tsuyahime (Yamagata) B**	**6.1 ± 0.0 c**	**258.0 ± 1.1 f**
**Tsuyahime (Shimane)**	**6.2 ± 0.1 c**	**298.0 ± 1.0 c**
**Tsuyahime (Miyagii)**	**5.3 ± 0.1 d**	**304.0 ± 0.8 c**
**Koshihikari (Saga)**	**6.3 ± 0.0 b**	**352.0 ± 1.1 a**
**Koshihikari (Ibaraki) A**	**5.5 ± 0.1 c**	**279.0 ± 1.8 d**
**Koshihikari (Ibaraki) B**	**5.5 ± 0.0 c**	**272.0 ± 2.0 e**
**Koshihikari (Shimane)**	**5.9 ± 0.0 c**	**294.0 ± 1.6 c**
**Koshihikari (Niigata) A**	**6.4 ± 0.0 b**	**292.0 ± 0.5 c**
**Koshihikari (Niigata) B**	**6.4 ± 0.0 b**	**270.0 ± 1.1 e**
**Koshihikari (Yamagata) A**	**5.8 ± 0.0 c**	**273.0 ± 1.2 e**
**Koshihikari (Yamagata) B**	**5.6 ± 0.0 c**	**277.0 ± 0.5 d**
**Koshihikari (Ishikawa)**	**5.5 ± 0.1 c**	**272.0 ± 0.8 e**
**Koshihikari (Yamanashi)**	**5.1 ± 0.1 d**	**285.0 ± 1.6 d**
**Milkyqueen (Kyoto)**	**6.0 ± 0.0 c**	**287.0 ± 1.6 d**
**Milkyqueen (Yamagata)**	**5.2 ± 0.0 d**	**271.0 ± 0.5 e**
**Yumepirika (Hokkaidou) A**	**5.8 ± 0.0 c**	**311.0 ± 1.1 c**
**Yumepirika (Hokkaidou) B**	**6.4 ± 0.0 b**	**335.0 ± 0.8 b**

Different letters (a, b, c, d, e, f) indicate significant difference. Each values are expressed on dry basis.

**Table 3 foods-14-03097-t003:** Textural properties of boiled rice grains of 32 rice samples from 2022.

	Surface layer	Overall	Surface layer	Overall	BalanceA1	BalanceA2
	Hardness (H1)	Hardness (H2)	Stickiness (S1)	Stickiness (S2)		
	(N/cm^2^)	(N/cm^2^)	(N/cm^2^)	(N/cm^2^)	(A3/A1)	(A6/A4)
**Gohyakukawa**	**0.57 ± 0.22 c**	**17.50 ± 2.27c**	**−0.085 ± 0.040 a**	**−3.01 ± 0.63 b**	**0.32 ± 0.21 d**	**0.09 ± 0.02 d**
**Kazesayaka**	**0.61 ± 0.23 c**	**16.58 ± 2.27 c**	**−0.108 ± 0.061 b**	**−3.46 ± 0.77 c**	**0.38 ± 0.19 c**	**0.12 ± 0.04 c**
**Sasanishiki**	**0.65 ± 0.21 b**	**16.28 ± 2.44 c**	**−0.078 ± 0.048 a**	**−2.87 ± 0.78 b**	**0.19 ± 0.10 e**	**0.09 ± 0.04 d**
**Ginganoshizuku**	**0.55 ± 0.21 c**	**16.43 ± 2.07 c**	**−0.078 ± 0.051 a**	**−2.86 ± 0.89 b**	**0.31 ± 0.27 d**	**0.10 ± 0.05 d**
**Hatsushimo**	**0.57 ± 0.27 c**	**19.02 ± 1.15 a**	**−0.125 ± 0.047 b**	**−3.39 ± 0.57 c**	**0.44 ± 0.17 b**	**0.11 ± 0.04 d**
**Koshiibuki**	**0.69 ± 0.17 b**	**17.10 ± 2.06 c**	**−0.162 ± 0.074 d**	**−3.49 ± 0.55 c**	**0.38 ± 0.15 c**	**0.14 ± 0.04 b**
**Haenuki**	**0.47 ± 0.17 d**	**13.63 ± 1.77 e**	**−0.101 ± 0.037 b**	**−1.79 ± 0.34 a**	**0.48 ± 0.23 b**	**0.15 ± 0.03 b**
**Tsugaruroman**	**0.50 ± 0.17 d**	**13.66 ± 2.25 e**	**−0.105 ± 0.039 b**	**−1.91 ± 0.60 a**	**0.38 ± 0.16 c**	**0.13 ± 0.04 c**
**Aichinokaori**	**0.69 ± 0.19 b**	**18.52 ± 1.94 a**	**−0.132 ± 0.056 c**	**−3.84 ± 0.57 c**	**0.38 ± 0.19 c**	**0.12 ± 0.04 c**
**Yuudai21**	**0.46 ± 0.16 d**	**15.37 ± 2.10 d**	**−0.112 ± 0.055 b**	**−3.54 ± 0.49 c**	**0.61 ± 0.27 a**	**0.17 ± 0.04 a**
**Akitakomachi (Ibaraki)**	**0.67 ± 0.14 b**	**17.27 ± 1.16 c**	**−0.083 ± 0.045 a**	**−3.43 ± 0.54 c**	**0.20 ± 0.10 e**	**0.09 ± 0.04 d**
**Akitakomachi (Chiba)**	**0.79 ± 0.16 a**	**17.21 ± 1.18 c**	**−0.169 ± 0.051 d**	**−3.85 ± 0.44 e**	**0.33 ± 0.15 d**	**0.14 ± 0.04 b**
**Akitakomachi (Akita) A**	**0.77 ± 0.17 a**	**17.92 ± 1.49 b**	**−0.154 ± 0.051 c**	**−3.80 ± 0.43 e**	**0.38 ± 0.10 c**	**0.13 ± 0.04 c**
**Akitakomachi (Akita) B**	**0.82 ± 0.20 a**	**18.70 ± 1.70 a**	**−0.154 ± 0.062 c**	**−3.47 ± 0.40 c**	**0.29 ± 0.14 e**	**0.12 ± 0.04 c**
**Tsuyahime (Yamagata) A**	**0.79 ± 0.22 a**	**17.99 ± 1.53 b**	**−0.192 ± 0.067 e**	**−3.88 ± 0.32 e**	**0.37 ± 0.12 c**	**0.13 ± 0.03 c**
**Tsuyahime (Yamagata) B**	**0.89 ± 0.18 a**	**19.38 ± 1.36 a**	**−0.194 ± 0.066 e**	**−3.82 ± 0.40 d**	**0.33 ± 0.14 d**	**0.13 ± 0.04 c**
**Tsuyahime (Shimane)**	**0.76 ± 0.14 a**	**1749 ± 1.33 b**	**−0.188 ± 0.049 e**	**−3.68 ± 0.36 d**	**0.40 ± 0.13 c**	**0.15 ± 0.04 b**
**Tsuyahime (Miyagii)**	**0.84 ± 0.15 a**	**18.18 ± 1.77 a**	**−0.191 ± 0.054 e**	**−3.78 ± 0.54 d**	**0.38 ± 0.11 c**	**0.15 ± 0.04 b**
**Koshihikari (Saga)**	**0.63 ± 0.15 c**	**17.07 ± 1.41 c**	**−0.115 ± 0.046 b**	**−3.21 ± 0.56 c**	**0.32 ± 0.15 d**	**0.11 ± 0.03 d**
**Koshihikari (Ibaraki) A**	**0.49 ± 0.18 d**	**15.93 ± 1.99 d**	**−0.103 ± 0.045 b**	**−3.23 ± 0.41 c**	**0.43 ± 0.16 b**	**0.14 ± 0.05 b**
**Koshihikari (Ibaraki) B**	**0.78 ± 0.20 a**	**18.11 ± 1.34 a**	**−0.144 ± 0.043 c**	**−3.77 ± 0.45 d**	**0.32 ± 0.12 d**	**0.12 ± 0.03 c**
**Koshihikari (Shimane)**	**0.73 ± 0.21 b**	**18.17 ± 1.44 a**	**−0.152 ± 0.039 c**	**−3.78 ± 0.33 d**	**0.35 ± 0.14 d**	**0.11 ± 0.04 d**
**Koshihikari (Niigata) A**	**0.72 ± 0.17 b**	**17.67 ± 1.60 b**	**−0.138 ± 0.036 c**	**−3.86 ± 0.33 e**	**0.34 ± 0.13 d**	**0.12 ± 0.03 c**
**Koshihikari (Niigata) B**	**0.83 ± 0.21 a**	**19.38 ± 1.52 a**	**−0.162 ± 0.048 d**	**−3.50 ± 0.50 c**	**0.35 ± 0.12 d**	**0.12 ± 0.03 c**
**Koshihikari (Yamagata) A**	**0.76 ± 0.17 a**	**17.42 ± 1.21 c**	**−0.166 ± 0.049 d**	**−3.67 ± 0.32 d**	**0.36 ± 0.13 d**	**0.13 ± 0.04 c**
**Koshihikari (Yamagata) B**	**0.74 ± 0.19 b**	**18.26 ± 1.30 a**	**−0.166 ± 0.047 d**	**−3.76 ± 0.40 d**	**0.41 ± 0.14 c**	**0.13 ± 0.03 c**
**Koshihikari (Ishikawa)**	**0.66 ± 0.15 b**	**17.31 ± 1.32 c**	**−0.154 ± 0.037 c**	**−3.40 ± 0.43 c**	**0.42 ± 0.18 c**	**0.14 ± 0.04 b**
**Koshihikari (Yamanashi)**	**0.73 ± 0.16 b**	**18.56 ± 1.37 a**	**−0.159 ± 0.045 d**	**−3.37 ± 0.39 c**	**0.37 ± 0.14 d**	**0.12 ± 0.03 c**
**Milkyqueen (Kyoto)**	**0.69 ± 0.16 b**	**15.68 ± 1.91 d**	**−0.147 ± 0.064 c**	**−3.96 ± 0.44 e**	**0.42 ± 0.20 c**	**0.17 ± 0.05 a**
**Milkyqueen (Yamagata)**	**0.66 ± 0.15 b**	**15.85 ± 2.07 d**	**−0.143 ± 0.059 c**	**−3.94± 0.39 e**	**0.42 ± 0.17 c**	**0.16 ± 0.05 a**
**Yumepirika (Hokkaidou) A**	**0.81 ± 0.17 a**	**17.15 ± 1.80 c**	**−0.183 ± 0.085 e**	**−3.95 ± 0.35 e**	**0.39 ± 0.19 c**	**0.17 ± 0.05 a**
**Yumepirika (Hokkaidou) B**	**0.87 ± 0.14 a**	**18.76 ± 1.37 a**	**−0.206 ± 0.054 f**	**−4.61± 0.34 f**	**0.37 ± 0.09 d**	**0.15 ± 0.05 b**

Different letters (a, b, c, d, e, f) indicate significant difference.

**Table 4 foods-14-03097-t004:** Pasting properties of 32 unpolished rice samples from 2022.

	Max.vis	Mini.vis	BD	Fin.vis	Setb	Cons	Max/Fin
	(RVU)	(RVU)	(RVU)	(RVU)	(RVU)	(RVU)	
**Gohyakukawa**	**315.4 ± 1.2 c**	**118.5 ± 0.9 d**	**196.9 ± 0.2 e**	**238.3 ± 1.4 d**	**−77.1 ± 0.2 c**	**119.9 ± 0.4 b**	**1.32 ± 0.01 d**
**Kazesayaka**	**294.9 ± 3.2 c**	**112.2 ± 0.2 d**	**182.7 ± 3.4 e**	**223.4 ± 0.4 d**	**−71.5 ± 2.8 c**	**111.2 ± 0.5 b**	**1.32 ± 0.01 d**
**Sasanishiki**	**317.7 ± 2.5 c**	**120.1 ± 1.4 d**	**197.6 ± 1.2 e**	**234.7 ± 1.7 d**	**−83.1 ± 0.8 d**	**114.5 ± 0.4 b**	**1.35 ± 0.00 d**
**Ginganoshizuku**	**282.6 ± 0.6 d**	**122.7 ± 0.7 d**	**160.0 ± 1.4 g**	**239.9 ± 0.6 d**	**−42.7 ± 1.3 b**	**117.2 ± 0.1 b**	**1.18 ± 0.01 d**
**Hatsushimo**	**282.2 ± 4.1 d**	**108.3 ± 1.6 e**	**173.9 ± 2.5 f**	**232.2 ± 2.9 d**	**−50.0 ± 1.2 b**	**123.9 ± 1.4 b**	**1.22 ± 0.00 d**
**Koshiibuki**	**315.4 ± 16.1 c**	**110.7 ± 3.8 e**	**204.7 ± 12.3 e**	**206.8 ± 4.9 e**	**−108.6 ± 11.2 e**	**96.1 ± 1.1 c**	**1.53 ± 0.04 c**
**Haenuki**	**295.3 ± 2.8 c**	**113.1 ± 0.2 d**	**182.2 ± 3.1 e**	**248.5 ± 2.3 c**	**−76.8 ± 0.5 c**	**105.4 ± 2.5 b**	**1.31 ± 0.00 d**
**Tsugaruroman**	**271.5 ± 6.7 d**	**118.3 ± 1.9 d**	**153.1 ± 4.8 g**	**226.1 ± 3.2 d**	**−45.3 ± 3.4 b**	**107.8 ± 1.4 b**	**1.20 ± 0.01 d**
**Aichinokaori**	**303.0 ± 6.2 c**	**126.6 ± 1.9 d**	**176.4 ± 4.2 f**	**205.8 ± 2.4 e**	**−97.2± 3.8 e**	**79.2 ± 0.4 d**	**1.47 ± 0.01 c**
**Yuudai21**	**348.9 ± 5.5 b**	**98.2 ± 0.7 e**	**250.7 ± 4.8 b**	**191.5 ± 1.1 e**	**−157.3 ± 4.4 g**	**93.4 ± 0.4 c**	**1.82 ± 0.02 b**
**Akitakomachi (Ibaraki)**	**322.7 ± 0.6 c**	**129.3 ± 2.5 d**	**193.4 ± 1.8 e**	**251.9 ± 2.3 c**	**−70.8 ± 1.6 c**	**122.5 ± 0.2 b**	**1.28 ± 0.01 d**
**Akitakomachi (Chiba)**	**328.3 ± 0.3 c**	**131.4 ± 0.8 c**	**196.9 ± 0.5 e**	**251.1 ± 0.6 c**	**−77.2 ± 0.4 d**	**119.8 ± 0.1 b**	**1.31 ± 0.00 d**
**Akitakomachi (Akita) A**	**301.3 ± 8.4 c**	**126.4 ± 0.2 d**	**174.8 ± 8.6 f**	**264.0 ± 1.0 b**	**−37.2 ± 9.4 a**	**137.6 ± 0.8 a**	**1.14 ± 0.04 d**
**Akitakomachi (Akita) B**	**297.0 ± 1.2 c**	**123.2 ± 0.5 d**	**173.8 ± 0.8 f**	**260.6 ± 0.2 b**	**−36.3 ± 1.1 a**	**137.5 ± 0.3 a**	**1.14 ± 0.00 d**
**Tsuyahime (Yamagata) A**	**354.7 ± 6.5 b**	**143.9 ± 6.0 b**	**210.8 ± 0.6 d**	**269.9 ± 3.7 b**	**−84.8 ± 2.9 d**	**126.0 ± 2.3 d**	**1.31 ± 0.01 d**
**Tsuyahime (Yamagata) B**	**329.3 ± 2.5 c**	**138.6 ± 1.2 b**	**190.6 ± 3.7 e**	**281.9 ± 0.1 a**	**−47.3 ± 2.4 b**	**143.3 ± 1.4 a**	**1.17 ± 0.01 d**
**Tsuyahime (Shimane)**	**401.5 ± 4.9 a**	**151.4 ± 0.4 a**	**250.1 ± 5.2 b**	**287.4 ± 0.4 a**	**−114.1 ± 5.2 e**	**136.0 ± 0.0 a**	**1.40 ± 0.02 c**
**Tsuyahime (Miyagii)**	**342.4 ± 1.2 b**	**149.3 ± 0.9 a**	**193.0 ± 0.3 e**	**277.1 ± 0.9 a**	**−65.3 ± 0.3 c**	**127.8 ± 0.0 b**	**1.24 ± 0.00 d**
**Koshihikari (Saga)**	**362.0 ± 1.4 b**	**133.9 ± 1.9 c**	**228.1± 0.5 c**	**248.9 ± 2.7 c**	**−113.2 ± 1.3 e**	**115.0 ± 0.8 b**	**1.45 ± 0.01 d**
**Koshihikari (Ibaraki) A**	**351.5 ± 5.8 b**	**120.7 ± 3.1 d**	**230.8 ± 2.5 c**	**227.7 ± 2.6 d**	**−123.8 ± 3.2 f**	**107.0 ± 0.5 b**	**1.54 ± 0.01 c**
**Koshihikari (Ibaraki) B**	**383.3 ± 5.8 a**	**125.2 ± 1.2 d**	**258.1 ± 4.6 b**	**235.7 ± 1.1 d**	**−147.5 ± 4.7 g**	**110.5 ± 0.1 b**	**1.63 ± 0.02 b**
**Koshihikari (Shimane)**	**390.2 ± 1.8 a**	**137.0 ± 1.3 c**	**253.2 ± 0.5 b**	**251.6 ± 1.2 c**	**−138.6 ± 0.6 g**	**114.6 ± 0.1 b**	**1.55 ± 0.00 c**
**Koshihikari (Niigata) A**	**376.8 ± 0.6 b**	**136.0 ± 0.8 c**	**240.8 ± 0.1 b**	**253.5 ± 0.8 c**	**−123.3 ± 0.1 f**	**117.5 ± 0.0 b**	**1.49 ± 0.00 c**
**Koshihikari (Niigata) B**	**342.2 ± 5.6 b**	**126.3 ± 1.1 d**	**215.9 ± 4.5 d**	**247.7 ± 0.9 c**	**−94.5 ± 4.7 e**	**121.3 ± 0.1 b**	**1.38 ± 0.02 d**
**Koshihikari (Yamagata) A**	**344.1 ± 6.2 b**	**134.5 ± 1.7 c**	**209.6 ± 4.5 d**	**261.0 ± 0.8 b**	**−83.2 ± 5.4 d**	**126.4 ± 0.9 b**	**1.32 ± 0.02 d**
**Koshihikari (Yamagata) B**	**330.8 ± 0.8 c**	**140.4 ± 0.7 b**	**190.3 ± 0.1 e**	**281.1 ± 1.2 a**	**−49.6 ± 0.4 b**	**140.7 ± 0.5 a**	**1.18 ± 0.00 d**
**Koshihikari (Ishikawa)**	**359.0 ± 0.1 b**	**130.6 ± 1.7 c**	**228.4 ± 1.8 c**	**261.1 ± 1.0 b**	**−97.9 ± 1.1 e**	**130.5 ± 0.7 a**	**1.37 ± 0.01 d**
**Koshihikari (Yamanashi)**	**332.6 ± 2.1 c**	**130.3 ± 1.3 c**	**202.3 ± 0.8 e**	**253.2 ± 1.4 c**	**−79.4 ± 0.7 d**	**122.9 ± 0.1 b**	**1.31 ± 0.00 d**
**Milkyqueen (Kyoto)**	**392.7 ± 1.1 a**	**108.5 ± 1.1 e**	**284.2 ± 2.2 a**	**179.6 ± 0.9 e**	**−213.0 ±1.9 h**	**71.2 ± 0.2 d**	**2.19 ± 0.02 a**
**Milkyqueen (Yamagata)**	**335.6 ± 2.1 c**	**106.0 ± 0.5 e**	**229.6 ± 1.6 c**	**189.4 ± 0.5 e**	**−146.2 ± 1.6 g**	**83.4 ± 0.1 d**	**1.77 ± 0.01 b**
**Yumepirika (Hokkaidou) A**	**330.5 ± 1.5 c**	**137.0 ± 1.9 b**	**193.4 ± 0.5 e**	**257.5 ± 1.7 b**	**−72.9 ± 0.2 c**	**120.5 ± 0.2 b**	**1.28 ± 0.00 d**
**Yumepirika (Hokkaidou) B**	**328.0 ± 3.7 c**	**134.8 ± 3.5 c**	**193.2 ± 0.2 e**	**255.0 ± 3.0 b**	**−73.1 ± 0.7 c**	**120.1 ± 0.5 b**	**1.29 ± 0.00 d**

Different letters (a, b, c, d, e, f, g, h) indicate significant differences.

**Table 5 foods-14-03097-t005:** Correlation between results of analysis of pasting properties and fatty acid compositions of 32 unpolished Japonica rice samples from 2022.

	Fatty Acid							
	Myristic	Palmitic	Palmitoleic	Stearic	Oleic	Linoleic	α-Linoleic	Arachidic
	(14:0) (%)	(16:0) (%)	(16:1) (%)	(18:0) (%)	(18:1) (%)	(18:2*n*−6) (%)	(18:3*n*−3) (%)	(20:0) (%)
**Max.vis**	**0.50 ****	**−0.34**	**0.42 ***	**0.63 ****	**0.20**	**−0.24**	**0.22**	**0.49 ****
**Min.vis**	**−0.09**	**0.08**	**0.08**	**0.30**	**0.05**	**−0.22**	**0.19**	**0.07**
**BD**	**0.58 ****	**−0.40 ***	**0.42 ***	**0.56 ****	**0.19**	**−0.17**	**0.16**	**0.50 ****
**Fin.vis**	**−0.13**	**0.21**	**0.00**	**0.24**	**0.01**	**−0.23**	**0.18**	**−0.06**
**SB**	**−0.53 ****	**0.42 ***	**−0.30**	**−0.39 ***	**−0.15**	**0.04**	**−0.07**	**−0.43 ***
**Pt**	**0.48 ****	**−0.46 ****	**0.52 ****	**0.68 ****	**0.46 ****	**−0.56 ****	**0.02**	**0.57 ****
**Cons**	**−0.20**	**0.25**	**0.06**	**0.16**	**0.00**	**−0.21**	**0.12**	**−0.10**
**Set/Cons**	**−0.45 ****	**0.39 ***	**−0.24**	**−0.21**	**−0.11**	**−0.04**	**0.00**	**−0.35**
**Max/Min**	**0.49 ****	**−0.36 ***	**0.31**	**0.26**	**0.12**	**0.00**	**0.02**	**0.34**
**Max/Fin**	**0.46 ****	**−0.39 ***	**0.27**	**0.23**	**0.12**	**0.02**	**0.02**	**0.36 ***

* Correlation is significant at 5% based on Tukey’s multiple comparison method. ** Correlation is significant at 1% based on Tukey’s multiple comparison method.

**Table 6 foods-14-03097-t006:** Pasting properties of 21 unpolished rice samples in 2023.

	Max.vis	Min.vis	BD	Fin.vis	Setb	Cons	Max/Fin
	(RVU)	(RVU)	(RVU)	(RVU)	(RVU)	(RVU)	
**Koshihikari (Saga)**	**351.0 ± 4.5 b**	**130.1 ± 2.1 c**	**221.6± 2.5 c**	**240.4 ± 3.5 c**	**−111.3 ± 1.1 e**	**110.3 ± 1.4 b**	**1.5 ± 0.0 b**
**Koshihikari (Ibaraki)**	**357.0 ± 8.5 a**	**122.8 ± 2.5 c**	**234.2 ± 6.0 c**	**228.6 ± 3.4 c**	**−128.4 ± 5.1 f**	**105.8 ± 0.9 c**	**1.5 ± 0.0 b**
**Koshihikari (Shimane)**	**364.5 ± 4.8 a**	**124.3 ± 1.2 c**	**240.2 ± 3.6 b**	**236.4 ± 0.9 c**	**−128.1 ± 3.9 f**	**112.1 ± 0.3 b**	**1.5 ± 0.0 b**
**Koshihikari (Niigata) A**	**342.1 ± 0.5 c**	**122.0 ± 2.3 c**	**250.1 ± 2.8 a**	**231.1 ± 2.2 c**	**−111.0 ± 2.8 e**	**109.0 ± 0.1 b**	**1.5 ± 0.0 b**
**Koshihikari (Niigata) B**	**344.2 ± 2.9 c**	**123.3 ± 1.4 c**	**220.9 ± 1.6 c**	**232.9 ± 1.7 c**	**−111.3 ± 1.2 e**	**109.6 ± 0.4 b**	**1.5 ± 0.0 b**
**Koshihikari (Yamagata) A**	**358.2 ± 0.8 a**	**137.3 ± 0.6 b**	**220.9 ± 1.4 c**	**256.5 ± 2.2 b**	**−101.7 ± 2.9 d**	**119.1 ± 1.6 a**	**1.4 ± 0.0 c**
**Koshihikari (Yamagata) B**	**350.9 ± 0.2 b**	**123.4 ± 1.1 c**	**227.5 ± 0.9 c**	**230.0 ± 1.6 c**	**−120.8 ± 1.4 f**	**106.7 ± 0.5 c**	**1.5 ± 0.0 b**
**Milkyqueen (Kyoto)**	**363.9 ± 0.3 a**	**103.2 ± 0.5 e**	**260.7 ± 0.2 a**	**188.9 ± 0.2 d**	**−175.0 ± 0.1 h**	**85.7 ± 0.2 d**	**1.9 ± 0.0 a**
**Milkyqueen (Yamagata)**	**369.5 ± 0.6 a**	**103.6 ± 0.6 e**	**265.9 ± 1.2 a**	**195.0 ± 0.2 d**	**−174.5 ± 0.5 h**	**91.4 ± 0.8 d**	**1.9 ± 0.0 a**
**Tsuyahime (Yamagata) A**	**352.9 ± 2.8 b**	**142.7 ± 0.8 a**	**210.2 ± 2.0 d**	**264.8 ± 0.5 a**	**−88.1 ± 2.2 c**	**122.1 ± 0.2 a**	**1.3 ± 0.0 d**
**Tsuyahime (Shimane)**	**364.3 ± 1.9 a**	**150.0 ± 1.6 a**	**214.3 ± 0.4 d**	**269.4 ± 1.2 a**	**−94.9 ± 0.8 d**	**119.4 ± 0.4 a**	**1.4 ± 0.0 c**
**Yumepirika (Hokkaidou) A**	**324.5 ± 0.1 d**	**117.1 ± 1.5 d**	**207.4 ± 1.4 d**	**225.5 ± 2.5 c**	**−99.0 ± 2.4 d**	**108.3 ± 1.1 b**	**1.4 ± 0.0 c**
**Gohyakukawa**	**317.6 ± 3.1 e**	**124.7 ± 1.4 c**	**192.9 ± 1.8 e**	**250.9 ± 1.5 b**	**−66.8 ± 1.6 b**	**126.2 ± 0.1 a**	**1.3 ± 0.0 d**
**Kazesayaka**	**309.0 ± 1.0 e**	**118.1 ± 2.3 d**	**190.9 ± 1.3 e**	**235.2 ± 1.9 c**	**−73.8 ± 0.9 b**	**117.1 ± 0.4 a**	**1.3 ± 0.0 d**
**Sasanishiki**	**336.1 ± 2.8 c**	**126.5 ± 3.6 c**	**209.7 ± 0.8 d**	**247.0 ± 4.5 b**	**−89.1 ± 1.7 c**	**120.5 ± 0.9 a**	**1.4 ± 0.0 c**
**Ginganoshizuku**	**336.2 ± 1.6 c**	**129.1 ± 0.1 c**	**207.1 ± 1.5 d**	**252.5 ± 0.2 b**	**−83.7 ± 1.4 c**	**123.4 ± 0.1 a**	**1.3 ± 0.0 d**
**Hatsushimo**	**278.2 ± 0.6 f**	**114.0 ± 0.2 d**	**164.2 ± 0.8**	**244.4 ± 0.8 b**	**−33.8 ± 1.4 a**	**130.5 ± 0.6 a**	**1.1 ± 0.0 e**
**Koshiibuki**	**345.3 ± 1.4 b**	**116.5± 0.1 d**	**228.8 ± 1.4 c**	**217.7 ± 1.2 d**	**−127.6 ± 2.5 f**	**101.1 ± 1.1 c**	**1.6 ± 0.0 b**
**Haenuki**	**340.6 ± 3.2 c**	**119.1 ± 0.1 d**	**221.5 ± 3.1 c**	**230.0 ± 0.4 c**	**−110.6 ± 2.8 e**	**111.0 ± 0.3 b**	**1.5 ± 0.0 b**
**Tugaruroman**	**322.3 ± 2.5 d**	**124.5 ± 3.2 c**	**197.8 ± 0.7 e**	**138.0 ± 4.2 e**	**−84.3 ± 1.6 c**	**113.5 ± 0.9 b**	**1.4 ± 0.0 c**
**Yuudai21**	**358.3 ± 1.1 a**	**103.3 ± 0.0 e**	**254.9 ± 1.1 a**	**201.6 ± 0.1 d**	**−156.6 ± 1.0 g**	**98.3 ± 0.1 c**	**1.8 ± 0.0 a**

Different letters (a, b, c, d, e, f, g, h) indicate significant differences.

**Table 7 foods-14-03097-t007:** Fatty acid compositions of 32 unpolished rice samples from 2022.

	Palmitic Acid (%)	Stearic Acid (%)	Oleic Acid (%)	Linoleic Acid (%)	α-Linolenic Acid (%)	Arachidic Acid (%)
	(16:0)	(18:0)	(18:1)	(18:2*n*−6)	(18:3*n*−3)	(20:0)
**Gohyakukawa**	**22.4 ± 0.7 a**	**2.1 ± 0.0 a**	**36.4 ± 0.6 c**	**34.8 ± 0.4 a**	**1.4 ± 0.0 a**	**0.6 ± 0.0 b**
**Kazesayaka**	**22.9 ± 0.6 a**	**2.1 ± 0.0 a**	**35.9 ± 0.4 c**	**34.9 ± 0.5 a**	**1.3 ± 0.0 a**	**0.6 ± 0.0 b**
**Sasanishiki**	**22.2 ± 0.6 a**	**2.1 ± 0.0 a**	**38.0 ± 0.5 b**	**33.8 ± 0.4 b**	**1.1 ± 0.0 b**	**0.6 ± 0.0 b**
**Ginganoshizuku**	**23.4 ± 0.2 a**	**1.8 ± 0.0 b**	**34.9 ± 0.2 c**	**36.0 ± 0.2 a**	**1.2 ± 0.0 ab**	**0.5 ± 0.0 c**
**Hatsushimo**	**22.3 ± 0.3 a**	**2.1 ± 0.0 a**	**36.3 ± 0.3 c**	**35.4 ± 0.3 a**	**1.0 ± 0.0 b**	**0.5 ± 0.0 c**
**Koshiibuki**	**21.9 ± 0.2 b**	**2.1 ± 0.0 a**	**38.4 ± 0.2 b**	**33.5 ± 0.2 b**	**1.2 ± 0.0 ab**	**0.6 ± 0.0 b**
**Haenuki**	**22.7 ± 0.4 a**	**2.0 ± 0.0 a**	**36.4 ± 0.3 c**	**34.8 ± 0.2 a**	**1.3 ± 0.0 a**	**0.5 ± 0.0 c**
**Tsugaruroman**	**23.4 ± 0.4 a**	**1.5 ± 0.0 c**	**34.6 ± 0.2 c**	**36.3 ± 0.1 a**	**1.3 ± 0.0 a**	**0.4 ± 0.0 d**
**Aichinokaori**	**21.6 ± 0.6 b**	**1.7 ± 0.0 b**	**37.8 ± 0.4 b**	**35.0 ± 0.4 a**	**1.1 ± 0.0 b**	**0.5 ± 0.0 c**
**Yuudai21**	**22.0 ± 0.4 a**	**1.8 ± 0.0 b**	**36.9 ± 0.2 b**	**35.3 ± 0.2 a**	**1.2 ± 0.0 ab**	**0.5 ± 0.0 c**
**Akitakomachi (Ibaraki)**	**19.7 ± 0.0 b**	**2.2 ± 0.0 a**	**42.9 ± 0.0 a**	**30.9 ± 0.0 c**	**1.0 ± 0.0 b**	**0.7 ± 0.0 a**
**Akitakomachi (Chiba)**	**19.5 ± 0.1 b**	**2.3 ± 0.0 a**	**43.2 ± 0.1 a**	**30.5 ± 0.1 c**	**1.0 ± 0.0 b**	**0.7 ± 0.4 a**
**Akitakomachi (Akita) A**	**22.3 ± 0.1 a**	**1.9 ± 0.0 a**	**38.1 ± 0.1 b**	**33.6 ± 0.1 b**	**1.2 ± 0.0 ab**	**0.5 ± 0.0 c**
**Akitakomachi (Akita) B**	**22.5 ± 0.2 a**	**1.9 ± 0.0 a**	**37.8 ± 0.2 b**	**33.8 ± 0.2 b**	**1.2 ± 0.0 ab**	**0.5 ± 0.0 c**
**Tsuyahime (Yamagata) A**	**22.6 ± 0.2 a**	**2.2 ± 0.0 a**	**36.1 ± 0.2 c**	**34.9 ± 0.2 a**	**1.3 ± 0.0 a**	**0.5 ± 0.0 c**
**Tsuyahime (Yamagata) B**	**23.0 ± 0.3 a**	**2.0 ± 0.0 a**	**36.7 ± 0.3 b**	**34.2 ± 0.3 a**	**1.3 ± 0.0 a**	**0.5 ± 0.0 c**
**Tsuyahime (Shimane)**	**21.1 ± 0.6 b**	**2.5 ± 0.0 a**	**39.2 ± 0.6 a**	**32.8 ± 0.6 b**	**1.2 ± 0.0 ab**	**0.7 ± 0.0 a**
**Tsuyahime (Miyagi)**	**22.6 ± 0.3 a**	**2.1 ± 0.0 a**	**37.4 ± 0.3 b**	**33.7 ± 0.3 b**	**1.4 ± 0.0 a**	**0.5 ± 0.0 c**
**Koshihikari (Saga)**	**22.1 ± 2.1 a**	**2.4 ± 0.0 a**	**36.2 ± 1.1 c**	**34.8 ± 1.1 a**	**1.3 ± 0.0 a**	**0.7 ± 0.4 a**
**Koshihikari (Ibaraki) A**	**21.2 ± 1.0 b**	**2.1 ± 0.0 a**	**38.9 ± 0.5 a**	**33.5 ± 0.6 b**	**1.2 ± 0.0 ab**	**0.6 ± 0.0 b**
**Koshihikari (Ibaraki) B**	**21.2 ± 0.0 b**	**2.2 ± 0.0 a**	**38.9 ± 0.0 a**	**33.3 ± 0.0 b**	**1.3 ± 0.0 a**	**0.6 ± 0.0 b**
**Koshihikari (Shimane)**	**21.8 ± 0.0 b**	**2.4 ± 0.0 a**	**36.7 ± 0.0 b**	**34.6 ± 0.6 a**	**1.3 ± 0.0 a**	**0.6 ± 0.0 b**
**Koshihikari (Niigata) A**	**22.3 ± 0.2 a**	**1.9 ± 0.0 a**	**37.2 ± 0.2 b**	**34.5 ± 0.2 a**	**1.3 ± 0.0 a**	**0.5 ± 0.0 c**
**Koshihikari (Niigata) B**	**22.9 ± 1.3 a**	**1.9 ± 0.0 a**	**35.5 ± 0.3 c**	**35.3 ± 0.3 a**	**1.5 ± 0.4 a**	**0.5 ± 0.0 c**
**Koshihikari (Yamagata) A**	**23.7 ± 0.3 a**	**2.0 ± 0.0 a**	**34.9 ± 0.3 c**	**35.3 ± 0.3 a**	**1.3 ± 0.0 a**	**0.5 ± 0.0 c**
**Koshihikari (Yamagata) B**	**23.3 ± 0.3 a**	**2.0 ± 0.0 a**	**35.9 ± 0.3 c**	**34.7 ± 0.3 a**	**1.4 ± 0.0 a**	**0.5 ± 0.0 c**
**Koshihikari (Ishikawa)**	**22.5 ± 0.6 a**	**2.3 ± 0.0 a**	**37.1 ± 0.6 b**	**33.5 ± 0.3 b**	**1.3 ± 0.0 a**	**0.6 ± 0.0 b**
**Koshihikari (Yamanashi)**	**22.7 ± 1.5 a**	**2.1 ± 0.0 a**	**37.0 ± 0.5 b**	**33.9 ± 0.5 b**	**1.4 ± 0.0 a**	**0.6 ± 0.0 b**
**Milkyqueen (Kyoto)**	**20.6 ± 0.6 b**	**2.3 ± 0.0 a**	**37.9 ± 0.6 b**	**34.5 ± 0.6 a**	**1.2 ± 0.0 ab**	**0.7 ± 0.0 a**
**Milkyqueen (Yamagata)**	**22.9 ± 1.6 a**	**1.9 ± 0.0 a**	**35.6 ± 1.6 c**	**35.3 ± 1.6 a**	**1.4 ± 0.0 a**	**0.5 ± 0.0 c**
**Yumepirika (Hokkaido) A**	**24.0 ± 0.5 a**	**1.9 ± 0.0 a**	**34.6 ± 0.5 c**	**35.6 ± 0.3 a**	**1.2 ± 0.0 ab**	**0.5 ± 0.0 c**
**Yumepirika (Hokkaido) B**	**23.8 ± 0.3 a**	**1.8 ± 0.0 b**	**34.1 ± 0.3 c**	**36.1 ± 0.3 a**	**1.3 ± 0.0 a**	**0.5 ± 0.0 c**

Different letters (a, b, c, d) indicate significant differences.

**Table 8 foods-14-03097-t008:** Comparison of fatty acid compositions of 21 unpolished Japonica rice harvested in 2022 and 2023.

	Palmitic Acid (2022)	Palmitic Acid (2023)	Oleic Acid (2022)	Oleic Acid (2023)	Linoleic Acid (2022)	Linoleic Acid (2023)	α-Linoleic Acid (2022)	α-Linoleic Acid (2023)
	(16:0)	(16:0)	(18:1)	(18:1)	(18:2*n*−6)	(18:2*n*−6)	(18:3*n*−3)	(18:3*n*−3)
**Koshihikari (Saga)**	**22.1 ± 2.1 a**	**22.8 ± 0.1 a**	**36.2 ± 1.1 c**	**35.7 ± 0.1 b**	**34.8 ± 1.1 a**	**34.4 ± 0.1 a**	**1.3 ± 0.0 a**	**1.3 ± 0.0 b**
**Koshihikari (Ibaraki)**	**21.2 ± 1.0 b**	**21.2 ± 0.0 b**	**38.9 ± 0.5 a**	**39.3 ± 0.3 a**	**33.5 ± 0.6 b**	**32.4 ± 0.2 b**	**1.2 ± 0.0 ab**	**1.4 ± 0.0 ab**
**Koshihikari (Shimane)**	**21.8 ± 0.0 b**	**23.0 ± 0.0 a**	**36.7 ± 0.0 b**	**35.4 ± 0.0 b**	**34.6 ± 0.6 a**	**34.4 ± 0.2 a**	**1.3 ± 0.0 a**	**1.3 ± 0.0 b**
**Koshihikari (Niigata) A**	**22.3 ± 0.2 a**	**21.2 ± 0.2 b**	**37.2 ± 0.2 b**	**39.2 ± 0.2 a**	**34.5 ± 0.2 a**	**33.0 ± 0.2 a**	**1.3 ± 0.0 a**	**1.2 ± 0.0 c**
**Koshihikari (Niigata) B**	**22.9 ± 1.3 a**	**20.6 ± 0.3 b**	**35.5 ± 0.3 c**	**40.4 ± 0.3 a**	**35.3 ± 0.3 a**	**32.3 ± 0.2 b**	**1.5 ± 0.4 a**	**1.2 ± 0.0 c**
**Koshihikari (Yamagata) A**	**23.7 ± 0.3 a**	**21.6 ± 0.3 b**	**34.9 ± 0.3 c**	**38.8 ± 0.3 a**	**35.3 ± 0.3 a**	**33.2 ± 0.1 a**	**1.3 ± 0.0 a**	**1.1 ± 0.0 c**
**Koshihikari (Yamagata) B**	**23.3 ± 0.3 a**	**22.2 ± 0.3 b**	**35.9 ± 0.3 c**	**37.7 ± 0.3 b**	**34.7 ± 0.3 a**	**33.5 ± 0.2 a**	**1.4 ± 0.0 a**	**1.1 ± 0.0 c**
**Milkyqueen (Kyoto)**	**20.6 ± 0.6 b**	**21.2 ± 0.2 b**	**37.9 ± 0.6 b**	**37.8 ± 0.2 b**	**34.5 ± 0.6 a**	**33.9 ± 0.2 a**	**1.2 ± 0.0 ab**	**1.2 ± 0.0 c**
**Milkyqueen (Yamagata)**	**22.9 ± 1.6 a**	**21.5 ± 0.6 b**	**35.6 ± 1.6 c**	**38.3 ± 0.6 a**	**35.3 ± 1.6 a**	**33.4 ± 0.2 a**	**1.4 ± 0.0 a**	**1.2 ± 0.0 c**
**Tsuyahime (Yamagata) A**	**22.6 ± 0.2 a**	**21.5 ± 0.2 b**	**36.1 ± 0.2 c**	**38.4 ± 0.2 a**	**34.9 ± 0.2 a**	**33.6 ± 0.2 a**	**1.3 ± 0.0 a**	**1.1 ± 0.0 c**
**Tsuyahime (Shimane)**	**21.1 ± 0.6 b**	**22.2 ± 0.4 b**	**39.2 ± 0.6 a**	**38.9 ± 0.4 a**	**32.8 ± 0.6 b**	**31.8 ± 0.2 b**	**1.2 ± 0.0 ab**	**1.1 ± 0.0 c**
**Yumepirika (Hokkaidou) A**	**24.0 ± 0.5 a**	**23.7 ± 0.3 a**	**34.6 ± 0.5 c**	**36.3 ± 0.2 b**	**35.6 ± 0.3 a**	**33.8 ± 0.2 a**	**1.2 ± 0.0 ab**	**1.1 ± 0.0 c**
**Gohyakukawa**	**22.4 ± 0.7 a**	**22.8 ±0.2 a**	**36.4 ± 0.6 c**	**35.5 ± 0.2 b**	**34.8 ± 0.4 a**	**34.6 ± 0.1 a**	**1.4 ± 0.0 a**	**1.5 ± 0.0 a**
**Kazesayaka**	**22.9 ± 0.6 a**	**21.7 ± 0.3 b**	**35.9 ± 0.4 c**	**38.0 ± 0.3 a**	**34.9 ± 0.5 a**	**33.7 ± 0.2 a**	**1.3 ± 0.0 a**	**1.2 ± 0.0 c**
**Sasanishiki**	**22.2 ± 0.6 a**	**21.9 ± 0.4 b**	**38.0 ± 0.5 b**	**39.3 ± 0.2 a**	**33.8 ± 0.4 b**	**32.2 ± 0.1 b**	**1.1 ± 0.0 b**	**1.0 ± 0.0 d**
**Ginganoshizuku**	**23.4 ± 0.2 a**	**22.3 ± 0.2 a**	**34.9 ± 0.2 c**	**38.9 ± 0.2 a**	**36.0 ± 0.2 a**	**32.3 ± 0.2 b**	**1.2 ± 0.0 ab**	**1.0 ± 0.0 d**
**Hatsushimo**	**22.3 ± 0.3 a**	**23.0 ± 0.3 a**	**36.3 ± 0.3 c**	**35.9 ± 0.3 b**	**35.4 ± 0.3 a**	**34.9 ± 0.3 a**	**1.0 ± 0.0 b**	**1.0 ± 0.0 d**
**Koshiibuki**	**21.9 ± 0.2 b**	**21.8 ± 0.2 b**	**38.4 ± 0.2 b**	**36.7 ± 0.2 b**	**33.5 ± 0.2 b**	**34.8 ± 0.2 a**	**1.2 ± 0.0 ab**	**1.1 ± 0.0 c**
**Haenuki**	**22.7 ± 0.4 a**	**21.0 ± 0.3 b**	**36.4 ± 0.3 c**	**39.0 ± 0.3 a**	**34.8 ± 0.2 a**	**32.8 ± 0.2 b**	**1.3 ± 0.0 a**	**1.1 ± 0.0 c**
**Tugaruroman**	**23.4 ± 0.4 a**	**21.7 ± 0.2 b**	**34.6 ± 0.2 c**	**38.3 ± 0.2 a**	**36.3 ± 0.1 a**	**33.8 ± 0.1 a**	**1.3 ± 0.0 a**	**1.1 ± 0.0 c**
**Yuudai21**	**22.0 ± 0.4 a**	**21.8 ± 0.2 b**	**36.9 ± 0.2 b**	**39.2 ± 0.2 a**	**35.3 ± 0.2 a**	**32.9 ± 0.2 a**	**1.2 ± 0.0 ab**	**1.1 ± 0.0 c**

Different letters (a, b, c, d) indicate significant differences.

## Data Availability

The datasets generated for this study are available upon request to the corresponding author.

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
