# Peer review of "Change in Fatty Acid Composition in High-Temperature-Damaged Rice Grains and Its Effects on the Appearance and Physical Qualities of the Cooked Rice"

_foods, 2025, doi:10.3390/foods14173097_

Round 1
Reviewer 1 Report
Comments and Suggestions for Authors
The manuscript entitled 'Change in fatty acid composition in high temperature damaged rice grains and its effects to the appearance and physical qualities of the cooked rice' addresses the study using physicochemical profiling (RVA, GC, texture analyzer) and taste evaluation (Mido Meter) in Japonica rice exposed to natural temperature variations during 2022-2023 harvests. The manuscript is scientifically sound and represents a meaningful contribution to agricultural climate resilience research. In my opinion this manuscript is acceptable for publication after major revision addressing.
Comments to the Author:
Scientific Rigor:
Question 1:Insufficient representativeness of the sample, this study uses only 32 Japonica rice samples from Japan, limiting generalizability to other rice subspecies or global regions affected by climate change.
Question 2:The annual comparison samples do not match, the 2023 sample set (n=21) is not fully equivalent to 2022 (n=32), introducing bias in interannual comparison (Section 2.1).
Question 3:The detection method for fatty acids has not been verified, fatty acid analysis via GC lacks validation details (e.g., recovery rate, detection limit), undermining reliability (Section 2.12).
Question 4:The definitions of key indicators are ambiguous, "Mido value" calibration is not described; its correlation with sensory tests cites outdated sources (1992–2003) without recent validation (Section 2.9).
Introduction:
Question 5:There is insufficient elaboration on the research blank, fails to quantify the knowledge gap on fatty acids as palatability indicators under high temperature, citing only Taira (1983/1988).
Question 6:The theoretical mechanisms have not been connected, does not link starch synthase downregulation (cited from Mitsui 2013) to fatty acid changes, creating a mechanistic disconnect.
Question 7:The research objective is ambiguous, objectives are dispersed; unclear if focus is on damage mechanisms, evaluation methods, or mitigation strategies.
Methods:
Question 8:Storage at 5°C may alter lipid oxidation; no justification provided (Section 2.1).
Question 9:"Set/Cons" and "Max/Fin" indices misattribute amylopectin effects without referencing primary literature (Section 2.11).
Question 10:Tensipresser settings lack calibration details for hardness/stickiness metrics (Section 2.8).
Question 11:Sources for 2022 (27.5°C) and 2023 (30.6°C) temperature data are unreferenced (Abstract).
Question 12:No mention of replicates or batch controls in α-amylase assays (Section 2.10).
Question 13:The method for extracting fatty acids is too simple. Why was the n-hexane-KOH methanol extraction method used instead of modern methods such as Folch?
Novelty:
Question 14:Fatty acids lack innovation as a new indicator. Claiming fatty acids as "novel indicators" ignores prior work (e.g., Nakamura 2019, cited in Ref 40).
Question 15:It repeats the author's previous research. Overlaps with Ref 18/22/40 without clarifying incremental contributions (e.g., Mido Meter integration).
Question 16:The data increment in high-temperature years is limited. 2023 data merely confirms known trends from 2022, lacking mechanistic exploration.
Question 17:Ignore the influence of genetic background. Does not address genetic variation in fatty acid metabolism among cultivars.
Discussion
Question 18:Correlation ≠ causality. Assumes temperature directly causes fatty acid changes but provides no experimental evidence (e.g., controlled trials).
Question 19:Fails to discuss how oleic acid binds amylopectin long chains at molecular level (Section 3.6).
Question 20:“An excessive amount of n6 polyunsaturated fatty acid (PUFA) and a very high n6/n3 ratio promote the pathogenesis of several diseases.” Please add supporting references here.
Question 21:Excessive inference of health effects. n6/n3 ratio implications (Section 3.7) are speculative without human/animal data.
Question 22:When discussing the composition of fatty acids, the author failed to compare his own findings with Taira's (1983) classic research, especially neglecting an in-depth exploration of the differences in fatty acids between Indica rice and Japonica rice. All samples were japonica rice (Section 2.1), but the conclusion did not limit the subspecies range, which might mislead readers into believing that "fatty acid indicates high-temperature damage" applies to all rice.
Question 23:Practical application suggestions are missing. No proposals for mitigating high-temperature damage (e.g., breeding targets).
Conclusions
Question 24:Overstates fatty acids as "novel indicators" despite limited new evidence.
Question 25:Fails to answer how fatty acid changes mechanistically reduce palatability.
Question 26:Ignore the limitations of the method. Omits limitations of Mido Meter versus sensory panels.
Question 27:Chart quality issue. For example, Table 8 does not conform to the three-line table format, the font of the text in the table, the position of Note and so on. Figure 6 should be marked with Figures 6A and B. Check the chart issues throughout the manuscript to meet the requirements of the journal.
Question 28:Line 271: Writing error. “differencr” should be “difference”. It is necessary to check the entire manuscript for any writing error.
Question 29:Lacks mechanistic distinction from prior work. Fails to differentiate fatty acid-temperature mechanisms from Nakamura (2019, Ref 40), presenting incremental data as novel insight.
Question 30:References are outdated; incorporate recent studies (particularly last 5 years) to contextualize findings within current research.
Question 31:The Conclusion suggests emphasizing that "fatty acids as an indicator of high-temperature damage" have only been verified in japonica rice at present, and further research is needed in indica rice.
Abstract
Question 32: "oleic acid significantly increased ... which would lead to the lower quality of rice grains." is revised to "oleic acid significantly increased ... associated with lower rice grain quality.". The reason is "Would lead to" overstates causality; the study demonstrates correlation, not experimental causation.
Question 33: "fatty acid composition would become a novel and useful indicators" is revised to "fatty acid composition may serve as potential indicators".
Author Response
Answers to the comments by reviewer 1
The manuscript entitled 'Change in fatty acid composition in high temperature damaged rice grains and its effects to the appearance and physical qualities of the cooked rice' addresses the study using physicochemical profiling (RVA, GC, texture analyzer) and taste evaluation (Mido Meter) in Japonica rice exposed to natural temperature variations during 2022-2023 harvests. The manuscript is scientifically sound and represents a meaningful contribution to agricultural climate resilience research. In my opinion this manuscript is acceptable for publication after major revision addressing.
A; Thank you very much for your valuable and enlightening comments. We revised our manuscript according to the comments by reviewers. We would appreciate your kind review of our revised version again.
Question 1:Insufficient representativeness of the sample, this study uses only 32 Japonica rice samples from Japan, limiting generalizability to other rice subspecies or global regions affected by climate change.
A; Thank you for your meaningful question. We agree to your comment. Therefore, we added that “It should be considered that these results were only for our rice sample, 32 Japonica rice produced in Japan. It would be necessary to use more wide-range rice samples including Japonica rice from other countries, or Indica rice samples, if we would like to elaborate our conclusion more widely.” in L681-684.
Question 2:The annual comparison samples do not match, the 2023 sample set (n=21) is not fully equivalent to 2022 (n=32), introducing bias in interannual comparison (Section 2.1).
A; Thank you for your meaningful question. We agree to your comment. Therefore, we used 32 rice sample to evaluate qualities and find the important “quality indices”, such as Tase degree, fatty acid composition, Pt, etc. Then, we selected 21 rice samples, of which cultivars and areas are same as rice samples in 2023 from 32 samples in 2022, and compared the degree of effects on qualities by the temperatures between two years.
Question 3:The detection method for fatty acids has not been verified, fatty acid analysis via GC lacks validation details (e.g. recovery rate, detection limit), undermining reliability (Section 2.12).
A; Thank you for your important comment. According to your comment, we added explanation of the method from L276-280.
Question 4:The definitions of key indicators are ambiguous, "Mido value" calibration is not described; its correlation with sensory tests cites outdated sources (1992–2003) without recent validation (Section 2.9).
A; Thank you for your valuable comment. “Mido meter” was developed by the rice milling company, Toyo Rice, in 1990. Principle was clarified as described in Materials and Methods, but software was not clarified. Prof. Shoji and Prof.Kurasawa reported in an academic journal that it is useful for evaluation of rice palatability, in 1991 [36]. Since 1999 to 2001, our research project on rice palatability was adopted by Ministry of Agriculture, Forestry, and Fisheries, Japan, and we reported that correlation of “Mido value (taste degree)” was 0.72 for various Japanese Japonica rice cultivars [38]. Then many breeding centers adopted Mido meter for the selection of palatable rice cultivars. And Mido meter is, now, used widely in the rice market and rice competitions. Recently, Mido meter is used not only in Japan but also in China, Taiwan, and Korea. According to your comment, we added references [39-42] for Mido meter, for example, selection of palatable premium rice cultivars, utilization in business market, etc.
Introduction:
Question 5:There is insufficient elaboration on the research blank, fails to quantify the knowledge gap on fatty acids as palatability indicators under high temperature, citing only Taira (1983/1988).
A; Thank you for your valuable comment. We added other references about the research on the relation between lipid/fatty acid composition and ripening temperature by Kitta et al. [31]. And we added references about the relationship between lipid/fatty acid composition and eating quality [33-35] .
Question 6:The theoretical mechanisms have not been connected, does not link starch synthase downregulation (cited from Mitsui 2013) to fatty acid changes, creating a mechanistic disconnect.
A; Thank you for your valuable comment. We added two explanations, [10] and [11] in L39-42. The former shows that fatty acids bind with helix structure of amylose as a complex, and the latter shows that fatty acid composition changes by the chain length of amylopectin due to down-regulation of starch synthase.
Question 7:The research objective is ambiguous, objectives are dispersed; unclear if focus is on damage mechanisms, evaluation methods, or mitigation strategies.
A; Thank you for your valuable comment. Objectives of this research are (1)evaluation and (2) damage mechanism. In concrete, (1) to propose how to evaluate palatability of rice produced under high temperature year, 2022, and (2) to ascertain the effects by the high temperature on the rice quality. Therefore, we added abovementioned description in abstract. Furthermore, we added the explanation “Sato et al. [46] reported that the textural characteristic of Hardness/Adhesion ratio was useful as an index for damage in eating quality due to high-temperature during ripening, but amylose content was not suitable. Wakamatsu et al. [47] showed that the preferred protein content of unpolished rice was estimated to be 6.0 %~7.0 % considering the palatability because the occurrence rate of white-back kernels increased when protein content of unpolished rice was less than 6.0 %.” in L134-139.
Methods:
Question 8:Storage at 5°C may alter lipid oxidation; no justification provided (Section 2.1).
A; Thank you for your important comment. Qu et al. reported that rice quality is maintained well at 15℃ (doi.org/10.1016/j.foodchem.2024.142107). Therefore, we stored our rice samples at 5℃ and analyzed as soon as possible.
Question 9:"Set/Cons" and "Max/Fin" indices misattribute amylopectin effects without referencing primary literature (Section 2.11).
A; Thank you for your important comment. We cited our report as [58] for "Set/Cons" and "Max/Fin" indices.
Question 10:Tensipresser settings lack calibration details for hardness/stickiness metrics (Section 2.8).
A; Thank you for your important comment. As shown in Figure 1S, we used for balance degrees (= stickiness/hardness and adhesiveness/hardness), “BalanceH1”, S1/H1 (= ratio of surface stress), “BalanceH2”, S2/H2 (= ratio of overall stress), “BalanceA1”, A3/A1 (= ratio of surface work), and “BalanceA2”, A6/A4 (= ratio of work). These definitions were reported in our previous report.
Question 11:Sources for 2022 (27.5°C) and 2023 (30.6°C) temperature data are unreferenced (Abstract).
A; Thank you for your important comment. We added sources of temperature data in L155-157.
The average temperature during ripening was 30.6°C in 2023, on the contrary, average temperature was 27.5°C in 2022 according to according to AMeDAS (Automated Meteorological Data Acquisition System, Japan) [49] in L159-161.
Question 12:No mention of replicates or batch controls in α-amylase assays (Section 2.10).
A; Thank you for your comment. We added repetition numbers (= three times) in L254-255.
Question 13:The method for extracting fatty acids is too simple. Why was the n-hexane-KOH methanol extraction method used instead of modern methods such as Folch?
A; Thank you for your important comment. We used the same method which we reported in our previous paper (AOCS Official Method Ce2-66, 1997). We added it in L276-279.
This method is same with official method (AOCS Official Method Ce2-66, 1997) except the amount of KOH-MeOH solution (0.1 mL). By potassium hydroxide-methanol as reaction solution, 98.6 %-102 % recovery was confirmed in methyl-esterification for five kinds of standard triacylglycerols; tricaprylin, trilaurin, tripalmitin, tristearin, and triolein [59] in L276-280.
Novelty:
Question 14:Fatty acids lack innovation as a new indicator. Claiming fatty acids as "novel indicators" ignores prior work (e.g., Nakamura 2019, cited in Ref 40).
A; Thank you for your important comment. In our previous paper [11], we reported that high degree of amylopectin rice cultivars showed a high Pt, and a higher degree of unsaturated fatty acids and low CD of amylopectin rice showed a high linoleic acid content, while that of Pt and Cons showed a low value.
In this paper, we found the correlation between fatty acid contents and taste degree in case of high-temperature damaged rice samples. It would be innovative.
We added explanation in L414-417 as follows; (L414-428)
As shown in Figure 3, the following formula for estimating the balance A2 (A6/A4) (overall balance degree) , which was based on Fb3 (DP≧37), linoleic acid (18: 2n-6) and α- amylase activity, was obtained using 32 rice samples produced in 2022. The equation showed a multiple regression coefficient of 0.64**.
A2 (A6/A4) (overall balance degree) = - 0.016 × Fb3 (DP≧37) - 0.001 × linoleic acid - 0.110 × α- amylase activity
It seems that the long chains of amylopectin and α- amylase activities increased by high-temperature ripening, although palmitic acid, linoleic acid and α- linolenic acid decreased, which caused harder and non-sticky texture of the boiled rice grains.
And H2 (overall hardness) of boiled rice grains showed significant positive correlation with palmitoleic acid (16:1) (r = 0.45, p < 0.01), and S2 showed significant positive correlation with palmitoleic acid (16:1) (r = 0.44, p < 0.05). Moreover, L3 (surface adhesion) of boiled rice grains showed significant positive correlation with α- linolenic acid (r = 0.49, p < 0.01). As a result, it seemed that fatty acid is one of the indicators for taste degree of high temperature damaged rice.
Question 15:It repeats the author's previous research. Overlaps with Ref 18/22/40 without clarifying incremental contributions (e.g., Mido Meter integration).
A; Thank you for your valuable comments. According to your comment, we deleted [18] (L55-58). We deleted [22] and description L63-66.
Question 16:The data increment in high-temperature years is limited. 2023 data merely confirm known trends from 2022, lacking mechanistic exploration.
A; Thank you for your important comment. We agree to your comment. But, as we answer to your comment, objective of this report is to propose how to evaluate palatability of rice produced under high temperature year, 2022, and second objective is to ascertain the effects by the high temperature on the rice quality.
Therefore, we used 32 rice sample to evaluate qualities and found the important “quality indices”, such as Tase degree, fatty acid composition, Pt, etc. Then, we selected 21 rice samples, of which cultivars and areas are same as rice samples in 2023 from 32 samples in 2022, and compared the degree of effects on qualities using abovementioned main indices, Pt and fatty acid composition, by the temperatures between two years to ascertain the effects by the high temperature on the rice quality.
Question 17:Ignore the influence of genetic background. Does not address genetic variation in fatty acid metabolism among cultivars.
A; Thank you for your important comment. For genetic variation in fatty acid among various rice cultivars, we cited reference by Gofman et al. [30]. And according to your comment, we added another report by Kitta et al. [31] who analyzed lipid contents and fatty acid compositions using major non-glutinous rice cultivars from all over Japan for 4 years, and they concluded that crop year and temperature affect lipid contents and fatty acid composition. As you pointed out, variation in metabolism among cultivars is very important problem, therefore, we would like to continue our research including it as one of our tasks in the future.
Discussion
Question 18:Correlation ≠ causality. Assumes temperature directly causes fatty acid changes but provides no experimental evidence (e.g., controlled trials).
A; Thank you for your valuable comment. We agree to your comment. We think there are two ways. One is growing rice in an artificial weather chamber, phytotron, and analyzing fatty acid composition under different temperatures. Another way is to collect many rice samples cultivated in paddy fields under different temperatures. The former results are theoretical and more accurate, and the latter results are a little ambiguous but actual and real agricultural. In this research, we adopted the latter method. As we used 21 rice samples of the same cultivars and same growing areas, we can get meaningful comparison results through the statistical treatment although not perfect and ambiguous to some extent.
Question 19:Fails to discuss how oleic acid binds amylopectin long chains at molecular level (Section 3.6).
A; Thank you for your important comment. According to your comment, we added reference [10] on amylose-lipid complexes.
Figure 6 gives us the indirect proof fatty acids bind with starch and increase Pt.
Question 20:“An excessive amount of n6 polyunsaturated fatty acid (PUFA) and a very high n6/n3 ratio promote the pathogenesis of several diseases.”Please add supporting references here.
A; Thank you for your valuable comment. We added another review [80,81,82]. High n6/n3 fatty acids were reported to lead to cardiovascular disease, cancer, inflammatory, autoimmune diseases and other chronic diseases in L581-585.
Question 21:Excessive inference of health effects. n6/n3 ratio implications (Section 3.7) are speculative without human/animal data.
A; Thank you for your important comment. We added reference [80-82] in L664-666 which showed experimental data on the effects of n6/n3 fatty acids.
Question 22:When discussing the composition of fatty acids, the author failed to compare his own findings with Taira's (1983) classic research, especially neglecting an in-depth exploration of the differences in fatty acids between Indica rice and Japonica rice. All samples were japonica rice (Section 2.1), but the conclusion did not limit the subspecies range, which might mislead readers into believing that "fatty acid indicates high-temperature damage" applies to all rice.
A; Thank you for your important comment. According to your comment, we added more recent results about fatty acid composition by Goffman et al [30] in L85-88.
There are only a few reports on rice lipid, we measured fatty acid composition of Japanese rice in 2016 (ordinary temperature year) and reported in Cereal Chemistry in 2018. Then, we compared fatty acid composition with those in 2022 (high temperature) and 2023 (extremely high temperature). As shown in Figure below, oleic acid increased significantly, and linoleic acid decreased significantly. Therefore, we reported it in this paper.
And according to your comment, we added, in conclusion, “It should be considered that these results were only for our rice sample, 32 Japonica rice produced in Japan. It would be necessary to use more wide-range rice samples including Japonica rice from other countries, or Indica rice samples, if we would like to elaborate our conclusion more widely” in L681-684.
Question 23:Practical application suggestions are missing. No proposals for mitigating high-temperature damage (e.g., breeding targets).
A; Thank you for your kind comment. According to your comment, we added “These results would be useful for breeding high-temperature tolerant, palatable and healthy rice cultivars.” In L693-695.
Conclusions
Question 24:Overstates fatty acids as "novel indicators" despite limited new evidence.
A; Thank you for your important comment. In the field of science and technology on rice quality evaluation, starch structure and protein contents were considered as the main factors, and no research report was published about the relationship between fatty acids and taste of boiled rice. By the recent progress in the relationship between fatty acids and taste, lipid and fatty acid have been considered as 6th taste factor. We studied on the relationship between fatty acids and taste of boiled rice grins, and fortunately, significant correlation was shown, therefore, we think that fatty acid as “novel indicators for taste of boiled rice”.
Question 25:Fails to answer how fatty acid changes mechanistically reduce palatability.
A; Thank you for your important comment. Starch lipids are bound to starch granules in the form of amylose lipid complexes [10]. Particularly, amylose is one of the components of rice starch that greatly affects the quality and gelatinization properties of cooked rice. By binding with fatty acids, amylose-lipid complex changes pasting temperature and physical properties [11]. As oleic acid has only one unsaturated bond and molecule is similar with linear which tends to bind with amylose helix structure easily. But linoleic acid and linolenic acid have 2 or 3 unsaturated bonds, which makes fatty acid to bended structure. Therefore, it is more difficult for linoleic acid and linolenic acid to bind with amylose helix structure. We think that it would be the reason why oleic acid showed positive correlation and linoleic acid showed negative correlation with pasting temperature which bring about deterioration of physical properties of boiled rice grains.
Question 26:Ignore the limitations of the method. Omits limitations of Mido Meter versus sensory panels.
A; Thank you for your comment. We added about Mido Meter versus sensory panels in L116-124.
Question 27:Chart quality issue. For example, Table 8 does not conform to the three-line table format, the font of the text in the table, the position of Note and so on. Figure 6 should be marked with Figures 6A and B. Check the chart issues throughout the manuscript to meet the requirements of the journal.
A; Thank you for your important comment. We revised Table 8 and Figure 6.
Question 28:Line 325: Writing error. “differencr” should be “difference”. It is necessary to check the entire manuscript for any writing error.
A; Thank you for your important comment. We corrected the spelling of “difference”, and we also checked other part of our manuscript.
Question 29:Lacks mechanistic distinction from prior work. Fails to differentiate fatty acid-temperature mechanisms from Nakamura (2019, Ref 40), presenting incremental data as novel insight.
A; Thank you for your valuable comment. Our previous paper [11] reported fatty acid composition of rice grains produced in ordinary temperature year (2016). In abstract of this manuscript, we described that “Our second objective of this research was to ascertain the effects by the high temperature on the rice quality. It was shown that oleic acid significantly increased, and linoleic acid and palmitic acid decreased in 21 rice samples of same cultivars and production areas in 2023, an unusually hot year, as compared with those in 2022, ordinary hot year”. According to your comment, we added description, “Although we reported fatty acid composition of rice produced in 2016, ordinary temperature year [11], Table 8 shows the difference in fatty acid composition of rice produced in high temperature (2022) and extremely high temperature (2023).” in L645-649.
Question 30:References are outdated; incorporate recent studies (particularly last 5 years) to contextualize findings within current research.
A; Thank you for your valuable comment. According to your comment, we added references issued recently, such as [21], [24], [31], [38], [39], [40], [41], [42], [66], [76], [78], [79] and [82].
Question 31:The Conclusion suggests emphasizing that "fatty acids as an indicator of high-temperature damage" have only been verified in japonica rice at present, and further research is needed in indica rice.
A; Thank you for your important comment. According to your comment, we added “It should be considered that these results were only for our rice sample, 32 Japonica rice produced in Japan. It would be necessary to use more wide-range rice samples including Japonica rice from other countries, or Indica rice samples, if we would like to elaborate our conclusion more widely.” in conclusion L681-684.
Abstract
Question 32: "oleic acid significantly increased ... which would lead to the lower quality of rice grains." is revised to "oleic acid significantly increased ... associated with lower rice grain quality.". The reason is "Would lead to" overstates causality; the study demonstrates correlation, not experimental causation.
A; Thank you for your important comment. According to your comment, we revised to “It was shown that oleic acid significantly increased, and linoleic acid and palmitic acid decreased in 21 rice samples of same cultivars and production areas in 2023, an unusually hot year, as compared with those in 2022, ordinary hot year. In conclusion, both oleic acid contents and pasting temperatures were associated with lower rice grain quality.” in L22-26.
Question 33: "fatty acid composition would become a novel and useful indicators" is revised to "fatty acid composition may serve as potential indicators".
A; Thank you for your important comment. According to your comment, we revised to “fatty acid composition may serve as potential indicators for evaluation of rice palatability” in L20-21.
Reviewer 2 Report
Comments and Suggestions for Authors
Abstract
More relevant quantitative information is missing, as well as greater emphasis on the novel scientific contribution of the study.
Introduction
The introduction is lengthy and at times tangential. Repetitive content regarding high-temperature damage should be reduced, with a stronger focus on the scientific gaps—for example, “what is the gap that this study fills which previous studies have not addressed?” The hypothesis of the study should be stated more clearly.
Materials and Methods
How was the “degree of heat damage” in the grains determined?
Was the sampling from the years 2022 and 2023 truly paired? If so, this needs to be better explained.
Include an experimental flowchart to facilitate understanding of the sampling plan.
Incorporate a multivariate analysis (PCA or PLS) to integrate the multiple datasets (fatty acid composition, Mido value, RVA, texture, etc.).
3.1 Starch Composition
The analyses are detailed and coherent but there is an excess of tables and values that are not well integrated into the discussion. The authors should synthesize the results into a cluster analysis by cultivar or by intensity of thermal damage.
3.2 Protein and Phosphorus
Provide a better discussion of phosphorus as a marker of heat damage.
3.3 Textural Properties
Apply canonical correlation analysis or multiple regression to correlate texture with starch and lipid composition.
3.4 Mido Value (Taste Degree)
Include results or references from validated sensory tests using the Mido-meter to strengthen the interpretation.
3.7 Comparison between 2022 and 2023
Conduct appropriate significance tests (two-way ANOVA or MANOVA) to clearly assess the impact of temperature.
Final Considerations
I recommend that the authors summarize the practical implications of their findings and highlight the study’s limitations (e.g., absence of sensory validation) as well as future perspectives (e.g., applications in breeding programs, tests with Indica rice).
Author Response
Answers to the comments by reviewer 2
Abstract
More relevant quantitative information is missing, as well as greater emphasis on the novel scientific contribution of the study.
A: Thank you for your valuable comment. We revised abstract and added quantitative information, such as “And pasting temperatures (Pts) of polished rice flour showed significant positive correlation with surface hardness of boiled rice grains (r = 0.53, p<0.01), and significant negative correlation with their overall stickiness (r = -0.57, p<0.01). Furthermore, Pts showed significant positive correlations with oleic acid and negative correlations with linoleic acid” in L16-20.
Introduction
The introduction is lengthy and at times tangential. Repetitive content regarding high-temperature damage should be reduced, with a stronger focus on the scientific gaps—for example, “what is the gap that this study fills which previous studies have not addressed?” The hypothesis of the study should be stated more clearly.
A: Thank you very much for your valuable comment. We deleted L55-58, L63-66, L88-91, and L110-115. We added “Although we reported fatty acid composition of rice produced in 2016, ordinary temperature year [11], Table 8 shows the difference in fatty acid composition of rice produced in high temperature (2022) and extremely high temperature (2023).” in L646-649.
We described in abstract that our first objective is to propose how to evaluate palatability of rice produced in high temperature year, 2022, and that our second objective of this research was to ascertain the effects by the high temperature on the rice quality.
We added “Our hypothesis was that fatty acid composition changed depending on the ripening temperature and fatty acid composition affected eating quality of boiled rice” in introduction (L145-146).
Materials and Methods
How was the “degree of heat damage” in the grains determined?
A; Usually, ratio of chalky rice grains among all the grains is used as an indicator for degree of heat damage. In our previous reports, we proposed (1) acceleration of enzyme activities, (2) changes in pasting properties, (3) lowering of amylose contents as candidates for degree of heat damage.
Was the sampling from the years 2022 and 2023 truly paired? If so, this needs to be better explained.
A; As we showed in our manuscript, we used 32 rice samples in 2022 for evaluation of heat damaged rice, and we found that fatty acid composition and pasting properties revealed characteristic differences in rice grains produced in 2022. Then, we chose 21 rice samples of same cultivars and same producing regions in 2022 with those in 2023 and compared those between 2022 and 2023.
Include an experimental flowchart to facilitate understanding of the sampling plan.
A; We made a flowchart as below according to your comment.
Incorporate a multivariate analysis (PCA or PLS) to integrate the multiple datasets (fatty acid composition, Mido value, RVA, texture, etc.).
A; Thank you for your valuable comment. According to your comment, we carried out PCA for 32 samples in 2023 using all the results of physicochemical measurements and added as Figure 8 (L615-625).
3.1 Starch Composition
The analyses are detailed and coherent but there is an excess of tables and values that are not well integrated into the discussion. The authors should synthesize the results into a cluster analysis by cultivar or by intensity of thermal damage.
A; Thank you for your valuable comment. According to your comment, we carried out cluster analysis for 32 samples in 2023 using Pt, taste degree, oleic acid, linoleic acid, palmitic acid and Fb3(37>DP) as variables. We added this result as a Supplemental Figure 3.
3.2 Protein and Phosphorus
Provide a better discussion of phosphorus as a marker of heat damage.
A; Thank you for your valuable comments. We added explanation about phosphorus in L345-346.
Average phosphorus contents of ordinary Japonica rice were 285.0 ± 9.5 mg/100g, and they were diversified according to the cultivars and producing areas. As a result of statistical analysis, phosphorus contents showed significant positive correlation with long-chain fatty acids, such as arachidonic acid, icosenoic acid. A part of phosphorus is reported to exist as a phospholipid [66], role of phosphorus in ripening under high temperature should be studied in the future. In the previous study, the phosphorus contents showed significant correlation with sunlight hours [67]. In this study, phosphorus contents showed significant negative correlation with taste degree. As high temperature ripening deteriorates rice quality, this result seems to be harmonized with the report by Nakata et al [70].
3.3 Textural Properties
Apply canonical correlation analysis or multiple regression to correlate texture with starch and lipid composition.
A; Thank you for your valuable comments. According to your comment, we carried out multiple regression analysis and added the result in L414-430.
A2 (A6/A4) (overall balance degree) = - 0.016 × Fb3 (DP≧37) - 0.001 × linoleic acid - 0.110 × α- amylase activity
3.4 Mido Value (Taste Degree)
Include results or references from validated sensory tests using the Mido-meter to strengthen the interpretation.
A; Thank you for your valuable comment. We added references about Mido-meter from [34] to [41] in introduction L116-124.
3.7 Comparison between 2022 and 2023
Conduct appropriate significance tests (two-way ANOVA or MANOVA) to clearly assess the impact of temperature.
A; Thank you for your valuable comments.
We showed that “Table 8. shows that comparison of fatty acid compositions of 21 unpolished rice between harvested in 2022 and 2023. The ripening temperature in 2023 was 3 °C higher than in 2022. As a result, Oleic acid (38.0 ± 1.5) % in 2023 showed significantly higher than that of (36.5 ± 1.4) % in 2022 at the level of 1%. In contrast, Linoleic acid (33.4 ± 0.9) % and α-linoleic acid (1.2 ± 0.1) % in 2023 showed significantly lower than those of (34.8 ± 0.8) % and (1.3 ± 0.1) % in 2022 at the level of 1% respectively, and Palmitic acid (21.9 ± 0.8) % in 2023 showed a similar tendency that of (22.5 ± 0.9) % in 2022 at the levelof 5 %” in L640-645.
Final Considerations
I recommend that the authors summarize the practical implications of their findings and highlight the study’s limitations (e.g., absence of sensory validation) as well as future perspectives (e.g., applications in breeding programs, tests with Indica rice).
A; Thank you for your valuable comment. According to your comment, we added descriptions as below in conclusion.
“It should be considered that these results were only for our rice sample, 32 Japonica rice produced in Japan. It would be necessary to use more wide-range rice samples including Japonica rice from other countries, or Indica rice samples, if we would like to elaborate our conclusion more widely” in L682-685.
“Although our results lack sensory evaluation, this result would be useful for breeding high-temperature tolerant, and palatable and healthy rice cultivars.” in L694-696.
Reviewer 3 Report
Comments and Suggestions for Authors
The manuscript discusses the impact of cooking and thermal heating on fatty acid profile of 21 rice varieties and their sensorial characteristics and qualities. The authors performed detailed analytical tests on rice samples harvested in two years. The topic of the work should be of interest to the readership of the journal. However, the manuscript should be extensively revised as some referenced information and tables are missing in the manuscript file and supplementary information. The work can additionally benefit from an inclusion of additional tables for increasing the clarity of the work to the reader. In some sections, additional discussions should be provided to strengthen the work.
I have the following comments/suggestions to the authors:
- In lines 37-39 of the Introduction section, add additional information to the main text on how high temperatures can change starch structure in chalky rice grains, e.g. the amylopectin or and its chain length.
Please see a relevant publication below:
https://doi.org/10.1016/j.rsci.2022.03.002
- In line 79 of the Introduction section, “Taira reported that fatty acid composition is different between Indica rice and Japonica rice [30].”
Add additional information to the main text on the reason for the observed behavior by Taira et al. [30].
- In line 88, what do authors mean by “orfactory system”? please check and correct.
- In lines 89-91, bring examples of bitter, sour, or umami fatty acids.
- In section 2.2, add additional information on how moisture was measured. By weight difference of the samples?
- In section 2.4, add details of the followed protocol. Mentioning only the name of the method and a reference does not provide the reader with the sufficient information.
- In section 3.1, line 246, please also add and discuss the findings of Patindol and wang [30] and Cuili et al. https://doi.org/10.1016/j.rsci.2022.03.002 in this section.
- Add to the caption of figure 1 what do subparts A and B show.
- In line 307, “Protein content showed significant negative correlations with alpha-linoleic acid”
Provide additional discussion for the observed behaviour.
- In line 326, Table S2 is missing in the supplementary information and should be added. In line 392 Table S3 is missing in the supplementary information and should be added.
- Add an additional table to the supplementary information and present correlation values r and significance values p between H1, H2, S1, and S2.
- In lines 351-353, “It seems that the long chains of amylopectins increased by high-temperature ripening, which caused harder and non-sticky texture of the boiled rice grains”.
Compare your findings with that of Patindol and wang; Asaoka; or Cuili et al. and add additional discussions to the main text.
- In section 3.5, “The chalky rice grains showed of alpha- amylase activities showed higher than”
This part of the sentence does not read well and should be rewritten.
- In line 438, do authors intend to refer to eicosenoic acid? do authors intend to refer to lignoceric acid?
- Referring to lines 444-446, please explain if the long chain of amylopectin binds with palmitic acid, why a significant negative correlation between Pt and palmitic acid was obtained. Does the bound palmitic acid increases the Pt value? Did authors intend to refer to arachidic acid instead?
- In lines 470-471, “It seems that the Pt, long chains in amylopectin and oleic acid were increased by high-temperature ripening.”
Add a brief reasoning for the observed behavior.
- In line 485, provide an additional reference in addition to reference [67].
- Add the comparison of the n6/n3 ratios for the years 2022 and 2023 and compare the two in figure 8. Currently only results of the year 2022 are shown.
- In section 4, line 551, what do authors mean here by "eating qualities"? the definition is not clear to the reader. Similarly, in line 559, “”taste degree”. Please rephrase them.
- In the conclusions section, add additional discussion based on the performed tests among 21 rice samples, which samples are proposed among others in terms of sensorial acceptability and health properties taking into account the composition of fatty acids.
- Several existing grammatical errors and typing mistakes should be corrected:
- In line 109, “We investigated about”
- In line 271, “differencr”
- In line 337, “a simply tendency”
- In line 347, “Blance”
- In line 389, “similer”
- In line 485, “breat”
- In line 536, “than tha”
- In the manuscript title, “effect to” should be corrected to “effect on”
Similarly check for the existing grammatical errors or typing mistakes throughout the work and correct them if necessary.

Several grammatical errors and typing mistakes exist in the manuscript, which are required to be corrected (please see my enclosed comments).
Author Response
Answer to the comments by reviewer 3
The manuscript discusses the impact of cooking and thermal heating on fatty acid profile of 21 rice varieties and their sensorial characteristics and qualities. The authors performed detailed analytical tests on rice samples harvested in two years. The topic of the work should be of interest to the readership of the journal. However, the manuscript should be extensively revised as some referenced information and tables are missing in the manuscript file and supplementary information. The work can additionally benefit from an inclusion of additional tables for increasing the clarity of the work to the reader. In some sections, additional discussions should be provided to strengthen the work.
A; Thank you for your enlightening review and valuable comments. According to your comments, we cited more references and revised introduction and discussion and corrected mistypes. We would be grateful if you would kindly review our revised version.
I have the following comments/suggestions to the authors:
In lines 37-39 of the Introduction section, add additional information to the main text on how high temperatures can change starch structure in chalky rice grains, e.g. the amylopectin or and its chain length.
Please see a relevant publication below:
https://doi.org/10.1016/j.rsci.2022.03.002
A; Thank you for your valuable comment. According to your comment, we added references issued recently, such as [21], [24], [31], [38], [39], [40], [41], [42], [66], [76], [78], [79] and [82].
In line 79 of the Introduction section, “Taira reported that fatty acid composition is different between Indica rice and Japonica rice [30].”
Add additional information to the main text on the reason for the observed behavior by Taira et al. [30].
A; Thank you for your comment. Taira reported that fatty acid composition is different between Indica rice and Japonica rice [28], and he also analyzed fatty acid composition of early-maturation rice, mid-maturation rice, and late-maturation rice and compared the results [29]. He discussed that temperature during ripening of rice grains affected fatty acid composition [29]. Therefore, we cited his report and added it in discussion about the relationship between the difference in fatty acid composition for Indica and Japonica rice. On the other hand, Gofmann et al. reported that genetic divergence affected the lipid and fatty acid composition of rice bran oil [30]. Furthermore, we added another reference [31] as “Kitta et al. measured lipid contents and fatty acid composition of Japanese major non-glutinous rice cultivars for 4 years, and proposed that fatty acid composition was affected by the temperature during ripening [31]”.
In line 88, what do authors mean by “orfactory system”? please check and correct.
A; Thank you for your question. We are sorry that spell is not “orfactory” but “olfactory”. “Olfactory system” is one of the sensory system, particularly, about flavor. It mainly exists in the taste buds of tongue; For example, lingual lipase digests neutral fat to free fatty acids which are released to be perceived as the taste of triacylglycerol and to find nutritive lipids in food.
In lines 89-91, bring examples of bitter, sour, or umami fatty acids.
A; Thank you for your comment. Fatty acids have been proven to affect the eating quality of foods as one of the tasty substances in addition to 5 gustatory substances, such as sweet (such as sugar), bitter (such as quinine), sour (such as vinegar), salty (such as NaCl), and umami (such as sodium glutamate) substances; recently, fatty acid was proposed as a different independent tasty substances except abovementioned 5 types of tasty substances as written in L102-109.
In section 2.2, add additional information on how moisture was measured. By weight difference?
A; Thank you for your question. As you asked, we used oven dry method (difference of weight before and after drying at 135℃).
Calculation formula moisture content (%) = 100 x [(sample weight + aluminum cup weight)(before heating)-(sample weight + aluminum cup weight)(after heating)] / (sample weight before drying)
In section 2.4, add details of the followed protocol. Mentioning only the name of the method and a reference does not provide the reader with the sufficient information.
A; Thank you for your valuable comment. We added in L195-203. Starch granules were prepared from 7 various rice flours using the cold alkaline method. Each sample (4 g) was removed protein with 0.1 % sodium hydroxide (40 mL) in water bath with ice at 0°C for 3 h with stirring vigorously, and discarded the supernatant, and the precipitation was washed with distilled water until it became neutral. After this, each precipitation was removed fat with 60 % ethyl alcohol (40 mL) at 0°C for 0.5 h and discarded the supernatant, and the defatted precipitation was washed with acetone (40 mL) at 0°C for 0.5 h. After discarding the supernatant, residual starch granules were dried at room temperature.
In section 3.1, line 246, please also add and discuss the findings of Patindol and wang [30] and Cuili et al. https://doi.org/10.1016/j.rsci.2022.03.002 in this section.
A; Thank you for your important comment. We added it in L311-315.
Patindol and Wang [8] and Cuili et al. [64] reported that amylose content decreased and short chain of amylopectin increased in case of high temperature damaged rice. Our results were same as them for amylose, but, for amylopectin, long chains increased.
It would be due to the difference among the rice cultivars and growing district.
Add to the caption of figure 1 what do subparts A and B show.
A; Thank you for your question. We added as below.
(A); Relationship between AAC and λmax in high temperature ripening in 2022.
(B); Relationship between AAC and Fb3 (DP≧37 %) in high temperature ripening in 2022.(L346-347)
In line 307, “Protein content showed significant negative correlations with alpha-linolenic acid”
Provide additional discussion for the observed behavior.
A; Thank you for your comment. For the relationship between protein and α-linolenic acid, we would like to delete this sentence because other fatty acid did not show significant correlations and the reason is not clear now.
In line 326, Table S2 is missing in the supplementary information and should be added.
In line 392 Table S3 is missing in the supplementary information and should be added.
Add an additional table to the supplementary information and present correlation values r and significance values p between H1, H2, S1, and S2.
A; Thank you for your important comment. We were sorry for wrong table numbers. We corrected as below.
Table S2 was corrected to Table 3.
Table S3 was corrected to Table S1.
By the correction above, relationships among H1, H2, S1, and S2 are described in the main text (L426-431) as below.
And H2 (overall hardness) of boiled rice grains showed significant positive correlation with palmitoleic acid (16:1) (r = 0.45, p < 0.01), and S2 showed significant positive correlation with palmitoleic acid (16:1) (r = 0.44, p < 0.05). Moreover, L3 (surface adhesion) of boiled rice grains showed significant positive correlation with α- linolenic acid (r = 0.49, p < 0.01). As a result, it seemed that fatty acid is one of the indicators for taste degree of high temperature damaged rice.
In lines 351-353, “It seems that the long chains of amylopectins increased by high-temperature ripening, which caused harder and non-sticky texture of the boiled rice grains”.
Compare your findings with that of Patindol and wang; Asaoka; or Cuili et al. and add additional discussions to the main text.
A; Thank you for your important comment. Our answer is shown as below and we added it in the main text from L311-315.
Our results were consistent with Asaoka because amylose decreased for high-temperature ripened rice. Patindol and Wang [8] and Cuili et al. [64] reported that amylose content decreased and short chain of amylopectin increased in case of high temperature damaged rice. Our results were same as their results for amylose.
But, our results were different for amylopectin, because amylopectin long chains increased. The cause of the inconsistence in amylopectin may be due to the difference among the rice cultivars and growing districts.
In section 3.5, “The chalky rice grains showed of alpha- amylase activities showed higher than”
This part of the sentence does not read well and should be rewritten.
A; Thank you for your valuable comments. According to your comment, we rewrote as follows,
The α- amylase activities of chalky rice grains (L 473)
In line 438, do authors intend to refer to eicosenoic acid? do authors intend to refer to lignoceric acid?
A; Thank you for your valuable comment. According to your comment, we revised as follows,
a higher degree of USF (icoscenoic acid), and a higher degree of SF (arachidic acid, behenic acid, libnoceric acid) (L523-525)
Referring to lines 444-446, please explain if the long chain of amylopectin binds with palmitic acid, why a significant negative correlation between Pt and palmitic acid was obtained. Does the bound palmitic acid increases the Pt value? Did authors intend to refer to arachidic acid instead?
A; Thank you for your valuable comment. We added two explanations, [10] and [11] in L39-42. The former shows that fatty acids bind with helix structure of amylose as a complex, and the latter shows that fatty acid composition changes by the chain length of amylopectin due to down-regulation of starch synthase.
We are sorry that “palmitic acid” should be replaced with “arachidic acid” as you pointed out.
In lines 470-471, “It seems that the Pt, long chains in amylopectin and oleic acid were increased by high-temperature ripening.”
Add a brief reasoning for the observed behavior.
A; Thank you for your important comment. First of all, we added two explanations, [10] and [11] in L39-42. The former shows that fatty acids bind with helix structure of amylose as a complex, and the latter shows that fatty acid composition changes by the chain length of amylopectin due to down-regulation of starch synthase.
In binding of fatty acids with amylopectin long chains, single double bond structure of oleic acid is dominant compared with linoleic acid (2 double bonds) and linolenic acid (3 double bonds) because the molecular structure is not vended remarkably. Therefore, Pt of starch bound with oleic acid shows higher Pt than that of linoleic acid.
In line 485, provide an additional reference in addition to reference [67].
Arachidonic acid can be syntesized from linoleic acid, and is one of the major fatty acids in the brain, as well as DHA [67].
A; Thank you for your important comment. According to your comment, we added another reference below.
- McCloy, U.; Ryan, M.A.; Pencharz, P.B.; Ross, R.J.; Cunnane, S.C. A comparison of the metabolism of eighteen-carbon 13C-unsaturated fatty acids in healthy women. J Lipid Res. 2004; 45(3):474-85. doi: 10.1194/jlr.M300304-JLR200.
Add the comparison of the n6/n3 ratios for the years 2022 and 2023 and compare the two in figure 8. Currently only results of the year 2022 are shown.
A; Thank you for your important comment. According to your comment, we compared n6/n3 ratio between 2 years. Results are shown in the figure below.
In section 4, line 551, what do authors mean here by "eating qualities"? the definition is not clear to the reader. Similarly, in line 559, “taste degree”. Please rephrase them.
A; Thank you for your important comment. We revised introduction as below (L116-124).
“Mido meter” was developed by the rice milling company, Toyo Rice Co. Ltd., Japan in 1990. Shoji and Kurasawa [36] and Mizuta et al. [37] reported in academic journals that it is useful for evaluation of rice palatability, in 1991 and 1996. Since 1999 to 2001, our research project on rice palatability was adopted by Ministry of Agriculture, Forestry, and Fisheries, Japan, and we reported that correlation of “Mido value (taste degree)” was 0.72 for various Japanese Japonica rice cultivars [38]. Then many breeding centers adopted Mido meter for the selection of palatable rice cultivars [39,40]. And Mido meter is, now, used widely in the rice market and rice competitions [41,42]. Recently, Mido meter is used not only in Japan but also in China, Taiwan, and Korea [42,43].
In the conclusions section, add additional discussion based on the performed tests among 21 rice samples, which samples are proposed among others in terms of sensorial acceptability and health properties taking into account the composition of fatty acids.
A; Thank you for your valuable comment. According to your comment, we added descriptions as below in conclusion.
“It should be considered that these results were only for our rice sample, 32 Japonica rice produced in Japan. It would be necessary to use more wide-range rice samples including Japonica rice from other countries, or Indica rice samples, if we would like to elaborate our conclusion more widely” in L682-685.
“Although our results lack sensory evaluation, this result would be useful for breeding high-temperature tolerant, and palatable and healthy rice cultivars.” in L694-696.
Several existing grammatical errors and typing mistakes should be corrected:
A; Thank you very much for your important comment. We revised them and checked our manuscript again. And revised the title “to” to “on”.
In line 109, “We investigated about”
In line 271, “differencer”
In line 337, “a similarly tendency”
In line 347, “Balance”
In line 389, “similarer”
In line 485, “breast”
In line 536, “than thea”
In the manuscript title, “effect to” should be corrected to “effect on”
Similarly check for the existing grammatical errors or typing mistakes throughout the work and correct them if necessary.
A; Thank you for your valuable comment. We revised.
Round 2
Reviewer 1 Report
Comments and Suggestions for Authors
Thank you for revising and improving the manuscript. This is much clearer, but the following three Questions have not been resolved yet.
For Question 2:The annual comparison samples do not match, the 2023 sample set (n=21) is not fully equivalent to 2022 (n=32), introducing bias in interannual comparison (Section 2.1).
The author's answer did not solve the problem.
For Question 3:The detection method for fatty acids has not been verified, fatty acid analysis via GC lacks validation details (e.g. recovery rate, detection limit), undermining reliability (Section 2.12).
Detailed data needs to be added to the main text or Supplementary file.
Question 30:References are outdated; incorporate recent studies (particularly last 5 years) to contextualize findings within current research.
Newly added references are still outdated.
Author Response
Comments and Suggestions for Authors
Thank you for revising and improving the manuscript. This is much clearer, but the following three Questions have not been resolved yet.
A; Thank you very much for your kind and enlightening comments. According to your comments, we re-revised our manuscript. We would be grateful for your kind review of our re-revised manuscript.
For Question 2:The annual comparison samples do not match, the 2023 sample set (n=21) is not fully equivalent to 2022 (n=32), introducing bias in interannual comparison (Section 2.1).
The author's answer did not solve the problem.
A; Thank you very much for your important comment. Flowchart of our study is as follows;
We think there are two ways. One is growing rice in an artificial weather chamber, phytotron, and analyzing fatty acid composition under different temperatures. Another way is to collect many rice samples cultivated in paddy fields under different temperatures. The former results are theoretical and more accurate, and the latter results are a little ambiguous but actual and real agricultural. In this research, we adopted the latter method. As we used 21 rice samples of the same cultivars and same growing areas, we can get meaningful comparison results through the statistical treatment although not perfect and ambiguous to some extent.
For Question 3:The detection method for fatty acids has not been verified, fatty acid analysis via GC lacks validation details (e.g. recovery rate, detection limit), undermining reliability (Section 2.12).
Detailed data needs to be added to the main text or Supplementary file.
A; Thank you very much for your valuable comment. We revised our manuscript from L269-L272 and added data about recovery rate, detection limit and reproducibility coefficients of variation (CVR).
Brown rice flour (0.2 g) was added to 2 mL of hexane and mixed well and 2M potassium hydroxide−methanol solution (0.2 mL) was added. After centrifugation, the concentration of the supernatant was adjusted with hexane and subjected to fatty acid composition analyses by gas–liquid chromatography (GC) (Shimadzu model GC-9A gas chromatograph with capillary column and flame ionization detector (FID). This method is same with official method (AOCS Official Method Ce2-66, 1997) [59] except the amount of KOH-MeOH solution (0.1 mL). By potassium hydroxide-methanol as reaction solution, 98.6 %-102 % recovery was confirmed in methyl-esterification for five kinds of standard triacylglycerols; tricaprylin, trilaurin, tripalmitin, tristearin, and triolein. Reproducibility coefficients of variation (CVR) were low enough, ranged from 0.4 to 3.2 % for major fatty acids with peak area ratio, and those of minor fatty acids with peak area ratio were less than 1 %. And limit of detection was 0.01 g/100 g.
Question 30:References are outdated; incorporate recent studies (particularly last 5 years) to contextualize findings within current research.
Newly added references are still outdated.
A; Thank you very much for your enlightening comment. We added 7 new references as below.
- Mai Thi Phuong, N.; Nguyen Thi Thuy, L.; Tran Thi Anh, N.; Ngo Thi Hong, N.; Chu Thi Quynh, A.; To Thi Mai, H. Natural variation in fatty acid composition of diverse Vietnamese rice germplasm. Vietnam Journal of Biotechnology. 2023,21(1), 141-153.
- Jie, G.; Xinqiao, Z.; Dagang, C.; Ke, C.; Chanjuan, Y.; Juan, L.; Shaolong, L.; Youding, C.; Guorong, C.; Chuanguang, L. Effect of fat content on rice taste quality through transcriptome analysis. Genes. 2024, 15,81. doi. Org/10.3390/genes15010081.
- Liting, Z.; Yu, X.; Yage, D.; Tianyi, X.; Wenqiang, S.; Sibin, Y. Natural variation of fatty acid desaturase gene affects linolenic acid content and starch pasting viscosity in rice grains. Int. Mol. Sci. 2022. 23,12055. doi.org/10.3390/ijms230912055.
- Guan, L.; Zhang, M. Formation and release of cooked rice aroma. Cereal Sci. 2022. 103523. doi.org/10.1016/j.jcs.2022.103523.
- Itayagoshi S.; Ishibashi T.; Matsui T.; Hashimoto N.; Kasaneyama H.; Fukushima R. Relationship among sensory evaluation of eating quality, the gelatinization temperature, and the physical properties of highly palatable rice. Hokuriku Crop Sci. 2021, 56, 77-81.
- Antunes, MM.; Godoy, G.; Curi, R.; Visentainer, JV.; Bazotte, RB. The myristic acid: docosahexaenoic acid ratio versus the n-6 polyunsaturated fatty acid: n-3 polyunsaturated fatty acid ratio as nonalcoholic fatty liver disease biomarkers. Metab Syndr Relat Disord. 2022. 20(2), 69-78. Doi: 10.1089/met2021.0107.
- Ribas, FBT.; Gasparetto, H.; Salau, NPG. Rice bran oil valorization: A comprehensive review of minor compounds, extraction, advancements, and prospects. ACS Food Sci. Technol. 2025. 5, 877-897.
- Miura, E.; Takahashi, H.; Watanabe, A.; Ueda, K.; Kawamoto, T.; Sakurai, K.; Akagi, H. Pleiotropic effects of the rice qLTG3‑1 allele: enhancing low‑temperature germinability while reducing brown rice appearance quality. Euphytica 2024, 220:134 https://doi.org/10.1007/s10681-024-03388-1
Reviewer 2 Report
Comments and Suggestions for Authors
The current version of the manuscript represents a significant improvement over the previous one. The efforts made to enhance the overall quality of the work are evident, and the revisions have positively contributed to the clarity and robustness of the study.
Author Response
Thank you very much for your kind and enlightening review. As we re-revised our manuscript. We would be grateful if you would kindly review our manuscript.
Reviewer 3 Report
Comments and Suggestions for Authors
The authors revised the manuscript and responded to some of the provided comments. Although, modifications are considered in the manuscript, I do not find the quality of the work sufficient for publications as some of the responses to some of the comments show lack of sufficient knowledge of the authors on the subject or negligence of the authors in accurate interpretation of their results.
E.g. in lines 523-524, I requested that the authors correct the names of the compounds. Still, inaccurate names “icoscenoic acid” and “libnoceric acid” are used in the sentence after modification.
I requested that the authors provide examples of bitter, sour, or umami fatty acids. Instead of fatty acids, the authors brought NaCl or vinegar as examples.
In the conclusions section, I requested that the authors add a brief summary on which samples they propose based on the performed sensory analysis. In response, the authors mentioned in lines 693-695 that their results lack sensory evaluation.
The authors report analysis on sensory attributes such as hardness or stickiness of samples in Table 3, and taste degree based on the measured Mido values in Figure 4. Several analysis are performed to assess the sensory characteristics of the samples. The authors should be able to derive a conclusion based on the reported results.
In some sections, I requested additional explanations on the reported results, such as the relationship between protein and alpha-linolenic acid. The information is just removed by the authors without interpretation of the result with an inclusion of probable reasoning. Even mistakes existed in the interpretation of the results in other sections that are just corrected upon my request.
Still, mistakes exist in the sentence structure and in the newly added section, section 2.4, several grammatical errors exist, e.g. “each sample was removed protein” or “the precipitation were washed” or “each precipitation was removed” or “defatted precipitation was washed”, which are required to be corrected.
Taking into account the above, I cannot recommend further consideration of the work for peer-review or publication.
Comments on the Quality of English Languagegrammatical errors still exist in the work, which should be corrected (please see my comments).
Author Response
The authors revised the manuscript and responded to some of the provided comments. Although, modifications are considered in the manuscript, I do not find the quality of the work sufficient for publications as some of the responses to some of the comments show lack of sufficient knowledge of the authors on the subject or negligence of the authors in accurate interpretation of their results.
A; Thank you for your enlightening review. According to your comments, we re-revised our manuscript. We would appreciate if you would kindly review our manuscript, again.
E.g. in lines 523-524, I requested that the authors correct the names of the compounds. Still, inaccurate names “icoscenoic acid” and “libnoceric acid” are used in the sentence after modification.
A; We are very sorry not to correct the spellings of fatty acids. We revised in L519 and L520.
I requested that the authors provide examples of bitter, sour, or umami fatty acids. Instead of fatty acids, the authors brought NaCl or vinegar as examples.
A; As we wrote in our previous answer, taste of fatty acids is novel and independent sixth taste, and different from sweet, bitter, sour, salty, or umami taste. Therefore, it is impossible for us to classify and show the taste of fatty acids as one of well-known 5 tastes.
In the conclusions section, I requested that the authors add a brief summary on which samples they propose based on the performed sensory analysis. In response, the authors mentioned in lines 693-695 that their results lack sensory evaluation.
A; Thank you very much for your valuable comment. We added promising rice cultivars (Koshihikari and Tsuyahime) in conclusion in L690 -L691.
The authors report analysis on sensory attributes such as hardness or stickiness of samples in Table 3, and taste degree based on the measured Mido values in Figure 4. Several analysis are performed to assess the sensory characteristics of the samples. The authors should be able to derive a conclusion based on the reported results.
A; Thank you very much for your enlightening comment. We added a summary of our results, such as Table 3 and Figure 4 in conclusion in L687 to L689.
In some sections, I requested additional explanations on the reported results, such as the relationship between protein and alpha-linolenic acid. The information is just removed by the authors without interpretation of the result with an inclusion of probable reasoning. Even mistakes existed in the interpretation of the results in other sections that are just corrected upon my request.
A; Thank you for your valuable comment. Compared with other fatty acids, only linolenic acid showed significant correlation with protein content. Therefore, we could not reach the proper explanation for it. According to your comment, we restored the sentence “Protein content showed significant negative correlations with αーlinolenic acid (r = -0.42, p <0.05)” and added the explanation in L360 to L365.
Still, mistakes exist in the sentence structure and in the newly added section, section 2.4, several grammatical errors exist, e.g. “each sample was removed protein” or “the precipitation were washed” or “each precipitation was removed” or “defatted precipitation was washed”, which are required to be corrected.
A; Thank you for your valuable comment. We revised as follows in L186 to L192;
Starch granules were prepared from 7 various rice flours using the cold alkaline method [5341 50]. Starch granules were prepared from 7 various rice flours using the cold alkaline method. Each sample (4 g) was removed of protein with 0.1 % sodium hydroxide (40 mL) in water bath with ice at 0°C for 3 h by with stirring vigorously, and discarded the supernatant, and precipitate was precipitation were washed with distilled water until it becomes neutral. Thereafter After this, each precipitate precipitation was removed of fat with 60 % ethyl alcohol (40 mL) at 0°C for 0.5 h and supernatant was discarded the supernatant, and the defatted precipitate precipitation was washed with acetone (40 mL) at 0°C for 0.5 h. After discarding the supernatant, residual starch granules were dried at room temperature.
Taking into account the above, I cannot recommend further consideration of the work for peer-review or publication.
Comments on the Quality of English Language
grammatical errors still exist in the work, which should be corrected (please see my comments).